



# Terrestrial or marine? – Indications towards the origin of Ice Nucleating Particles during melt season in the European Arctic up to 83.7°N

Markus Hartmann[1], Xianda Gong[1,5], Simonas Kecorius[1], Manuela van Pinxteren[2], Teresa Vogl[1,3], André Welti[1,4], Heike Wex[1], Sebastian Zeppenfeld[2], Hartmut Herrmann[2], Alfred Wiedensohler[1], and Frank Stratmann[1]

[1]Experimental Aerosol and Cloud Microphysics, Leibniz Institute for Tropospheric Research, 04318, Leipzig, Germany
[2]Atmospheric Chemistry Department, Leibniz Institute for Tropospheric Research, 04318, Leipzig, Germany
[3]now at: Remote Sensing and The Arctic Climate System, Institute for Meteorology, University of Leipzig, 04103, Leipzig, Germany
[4]now at: Finnish Meteorological Institute, Helsinki, Finland
[5]now at: Center for Aerosol Science and Engineering, Department of Energy, Environmental and Chemical Engineering, Washington University in St. Louis, St. Louis, USA

**Correspondence:** Markus Hartmann (markus.hartmann@tropos.de)

**Abstract.** Ice nucleating particles (INPs) initiate the primary ice formation in clouds at temperatures above ca. -38°C and have an impact on precipitation formation, cloud optical properties and cloud persistence. Despite their roles in both weather and climate, INPs are not well characterized, especially in remote regions such as the Arctic. We present results from a ship-based campaign to the European Arctic in May to July 2017. We deployed a filter sampler and a continuous flow diffusion chamber
for off- and online INP analysis, respectively. We also investigated the ice nucleation properties of samples from different environmental compartments, i.e., the sea surface microlayer (SML), the bulk seawater (BSW), and fog water. Concentrations of INP ($N_{\mathrm{INP}}$) in the air vary between two to three orders of magnitudes at any particular temperature and are, except for the temperatures above -10°C and below -32°C, lower than in mid-latitudes. In these temperature ranges INP concentrations are the same or even higher than in the mid-latitudes. Heating of the filter samples to 95°C for 1 hour we found a significant reduction
in ice nucleation activity, i.e., indications that the INPs active at warmer temperatures are biogenic. At colder temperatures the INP population was likely dominated by mineral dust. The SML was found to be enriched in INP compared to the BSW in almost all samples. The enrichment factor (EF) varied mostly between 1 and 10, but EFs as high as 94.97 were also observed. Filtration of the seawater samples with 0.2 $\mu$m syringe filters lead to a significant reduction in ice activity, indicating the INPs are larger, and/or are associated with particles larger than 0.2 $\mu$m. A closure study showed that aerosolization of SML and/or
seawater alone cannot explain the observed air-borne $N_{\mathrm{INP}}$ unless significant enrichment of INP by a factor of $10^5$ takes place during the transfer from the ocean surface to the atmosphere. In the fog water samples with -3.47°C we observed the highest freezing onset of any sample. A closure study connecting $N_{\mathrm{INP}}$ in fog water and the ambient $N_{\mathrm{INP}}$ derived from the filter samples shows good agreement of the concentrations in both compartments, which indicates that INPs in the air are likely all activated into fog droplets during fog events. In a case study we considered a situation during which the ship was located in
the marginal sea ice zone and $N_{\mathrm{INP}}$ in air and the SML were highest in the temperature range above -10°C. Chlorophyll-a





measurements by satellite remote sensing point towards the waters in the investigated region being biologically active. Heat induced reduction of ice nucleating ability indicated the biogenic nature of the air-borne INPs. Similar slopes in the temperature spectra suggested a connection between the INP populations in the SML and the air. Air mass history had no influence on the observed air-borne INP population. Therefore, we conclude that during the case study collected air-borne INPs originated from
a local biogenic probably marine source.

## 1 Introduction

The Arctic is more sensitive to climate change than any other region on Earth, and changes are proceeding at an unprecedented pace and intensity (Serreze and Barry, 2011). The increase of the Arctic surface air temperature, the most prominent variable to indicate Arctic change, exceeds the warming in mid-latidutes by about 2 K (Wendisch et al., 2017; Overland et al., 2011;
Serreze and Barry, 2011). This enhanced warming phenomenon is referred to as Arctic Amplification (AA). Arctic peculiarities together with multiple feedback mechanisms are known to contribute to the enhanced sensitivity of the Arctic (Wendisch et al., 2017). And, while the individual processes are known, the relative contribution of each process, their strength and inter-linkage leading to AA are still a field of ongoing research (Serreze and Barry, 2011; Pithan and Mauritsen, 2014; Cohen et al., 2020; Wendisch et al., 2017).
The visible manifestation of the Arctic climate change is the perennial sea ice cover decline, which has intensified over the last decade (Lang et al., 2017; Kwok et al., 2009). The decline in sea-ice results in an overall increase in marine biological activity, which may give rise to new sources for aerosol particles and/or alter exisiting ones (Arrigo et al., 2008). Equivalently to the marine environment the thawing of the permafrost increases the terrestrial biological activity (Hinzman et al., 2005) and presumably the emission of primary aerosol particles and/or particle precursors (Creamean et al., 2020).
Aerosol particles are a key factor in cloud formation and can alter the microphysical properties of clouds (Pruppacher and Klett, 2010). The formation of clouds is further promoted by the increase in near surface water vapor concentration due to the extended open water areas. Clouds and their properties are essential for the energy budget of the Arctic boundary layer. Arctic clouds are often low level and they tend to warm the surface below the clouds (Intrieri, 2002; Shupe and Intrieri, 2004) and consequently cause more sea ice to melt (Vavrus et al., 2011). An increased cloud cover and the accompanying downward
longwave radiation also prevents new sea ice from growing again, reducing the sea ice cover in the following seasons (Liu and Key, 2014; Park et al., 2015).
A number of studies showed that mixed-phase clouds prevail, existing in the temperature range between 0 and -38°C, in the Arctic (e.g. Intrieri, 2002; Pinto, 1998; Shupe et al., 2006, 2011; Turner, 2005). These clouds, which are composed of a mixture of supercooled droplets and ice crystals, typically extend over large areas and display extraordinary longevity despite their
microphysically unstable nature. These clouds show a lower degree of glaciation in comparison to clouds at similar altitudes in other parts of the globe (Costa et al., 2017) which might be due to a lack of ice nucleating particles (INPs). INPs are the catalyst needed for the primary ice formation at temperatures relevant for mixed-phase clouds and thus essential to induce the freezing of supercooled liquid cloud droplets.



As INPs can directly affect the phase state of the cloud, their abundance and efficiency to initiate freezing also affects pre-
cipitation, life time and the radiative effects of clouds (e.g. Loewe et al., 2017; Prenni et al., 2007; Ovchinnikov et al., 2014;
Solomon et al., 2015). Solomon et al. (2018) even state that the influences of INPs regarding the radiative properties of Arctic
clouds are more important than those of CCN. This underlines the importance of gaining quantitative knowledge about the
abundance, properties, nature and sources of Arctic INPs.

Several previous studies have reported that marine as well as terrestrial sources contribute to Arctic INPs ice active at tem-
peratures above approximately -15°C. For the marine environment it was found that especially the sea surface microlayer
(SML) can be highly ice active (Alpert et al., 2011a, b; Bigg, 1996; Bigg and Leck, 2008; Irish et al., 2017, 2019; Knopf
et al., 2011; Leck and Bigg, 2005; Schnell and Vali, 1976; Wilson et al., 2015; Zeppenfeld et al., 2019). Especially marine
microorganism such as bacteria and algae as well as their exudates are thought to be the source for the INPs. Connections
to biological driven processes like plankton blooms have been made (Creamean et al., 2019). Another recent publication by
Kirpes et al. (2019) found locally produced open leads to be the dominant aerosol source in winter. The emitted sea spray
aerosol particles were found to possess organic coatings, consisting of marine saccharides, amino acids, fatty acids, and di-
valent cations. These substances are known from exopolymeric secretions produced by sea ice algae and bacteria, which, as
mentioned before are thought to be responsible for the ice activity in seawater. Studies on INPs at coastal sites tend to find
influences from marine and terrestrial sources, often with a contribution of biological INPs and seasonal changes (Creamean
et al., 2018; Šantl-Temkiv et al., 2019; Wex et al., 2019). For terrestrial sources, mineral dust itself is known to be relevant for
lower temperatures (Sanchez-Marroquin et al., 2020), but Tobo et al. (2019) showed for glacial outwash material that dust can
be the carrier for biological material, which is more ice active than the dust alone. Highly ice active biological INPs have also
been found Arctic ice cores from up to 500 years ago (Hartmann et al., 2019). Also millennia old permafrost soil was found
to contain biological INPs that can be mobilized into the atmosphere, lakes, rivers, and the ocean when the Permafrost thaws
(Creamean et al., 2020). This highlights the importance of biological INPs especially in the changing Arctic environment.

Despite past significant efforts and increase in knowledge, we still lack quantitative insights concerning the abundance, the
properties, and sources of Arctic INPs. Especially concerning the latter, the relative importance of marine vs. terrestrial sources
is still debated. Therefore, open questions addressed in this paper are

– What is the abundance of Arctic INPs and in what temperature range can they nucleate ice?

– What is the nature of Arctic INPs (biogenic material vs. mineral dust)?

– What is the origin of Arctic INPs (local vs. long range transport, marine vs. terrestrial)?

Thereby we will contribute to a better understanding concerning Arctic INPs and their potential effects on Arctic clouds.
Furthermore, we provide valuable data for evaluating and driving atmospheric models in a region which is still heavily under-
sampled.



## 2 Methods

### 2.1 Campaign overview

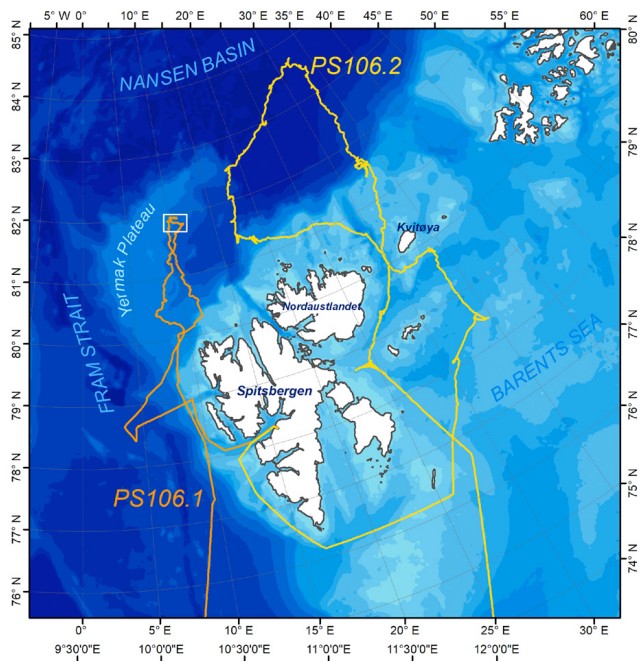

**Figure 1.** Overview of the main expedition area of the Polarstern cruises PS106.1 and PS106.2. Figure taken from Macke and Flores (2018).

The expedition PS106 of the research vessel Polarstern (Knust, 2017) was conducted between the end of May and mid-July 2017 in the Arctic Ocean (Wendisch et al., 2019). The measurements were performed as part of the PASCAL (Physical Feedbacks of Arctic Boundary Layer, Sea Ice, Cloud and Aerosol) campaign in the framework of the German Arctic Amplification: Climate Relevant Atmospheric and Surface Processes, and Feedback Mechanisms (AC)[3] project.

The first leg (PS106.1) started on 24 May in Bremerhaven (Germany) and ended 21 June in Longyearbyen (Svalbard) and featured a 10 day ice floe camp that was set up between 5 and 14 June 2017. The main area of investigation was the Arctic Ocean a few hundred kilometers northwest of Svalbard (see Fig. 1).

The expedition continued with its second leg (PS106.2) on 23 June from Lonyearbyen and ended 20 July in Tromsø (Norway). In comparison to PS106.1, the second leg focused on the area northeast of Svalbard, went up to higher latitudes (up to 83.7°N) and the vessel did not stop for extended stays at an ice floe.

As an overview about the meteorological situation during the campaign, Fig. S1 in the SI shows the frequency distributions for all meteorological parameters that were continuously measured on Polarstern. The mean and standard deviation of air temperature ($T_{air}$), relative humidity (RH) and atmospheric pressure (p) are given in the following: for the whole first leg are





$T_{air}$ = -0.01°C± 4.21°C, RH = 90.70 % ± 10.62 % and p = 1016.36 hPa ± 7.48 hPa, whereas the second leg the parameters were $T_{air}$ = 0.22°C± 2.71°C, RH = 94.82 % ± 6.09 % and p = 1006.84 hPa ± 5.12 hPa. During the time within the ice pack the averages of these parameters were as follows $T_{air}$ = -1,37°C± 1.50°C, RH = 94.35 % ± 4.54 % and p = 1011.27 hPa ± 8.52 hPa and out of the ice pack: $T_{air}$ = 4.75°C± 3.65°C, RH = 88.03 % ± 11.27 % and p = 1012.92 hPa ± 4.89 hPa.

For further details on the measurement strategy as well as the meteorological, sea ice, and cloud conditions during PASCAL refer to Wendisch et al. (2019) and the PS106 cruise report by Macke and Flores (2018).

### 2.2    Sample collection

In order to gain a comprehensive insight into the abundance and nature of INPs in the Arctic during summer time, samples from different compartments were taken. These included atmospheric, bulk seawater (BSW), sea surface microlayer (SML),

and fog samples. All samples were stored on the vessel directly after sampling in a cold room at -20°C and it was ensured that the samples stayed below 0°C during the transport to the Leibniz Institute for Tropospheric Research (TROPOS), were they were stored at -24°C until they were analyzed.

#### 2.2.1    Filter sampling

Aerosol particles were sampled using a low volume filter sampler (LVS; DPA14 SEQ LVS, DIGITEL Elektronik AG, Volketswil,

Switzerland) with a $PM_{10}$ inlet (DPM10/2.3/01, DIGITEL Elektronik AG, Volketswil, Switzerland). The sampler was located on top of a measurement container placed on the starboard side of the monkey island (ca. 30 m above sea level). It was operated with an average volumetric flow of 27.9 L min⁻¹. It should be noted that our flow rate is lower than the standardized flow rate for $PM_{10}$ inlets, hence our cut-off diameter is higher than 10 $\mu$m (ca. 11.7 $\mu$m). The LVS was routinely operated with an 8 hours sampling period, which results in a total sampled air volume of 13.4 m³ per filter sample. On four days the 8 hour cycle

was replaced by a 2 hour cycle to study possible diurnal variation. The filter sampler features sealed storage cassettes and an automated filter change that allows unsupervised sampling for multiple days. The samples were collected on polycarbonate pore filters (Nuclepore®, Whatman™; 0.2 $\mu$m pore size, 47 mm diameter). Usually 12 filters were prepared and put in place inside the sampler. Two field blanks were taken on each leg and were used to define the lower limit of observable $N_{INP}$. A list of the almost 200 filter samples can be found in Tab. S1 in the SI. The filter-derived $N_{INP}$ are also part of the overview of global

ship-borne INP measurements by Welti et al. (2020).

#### 2.2.2    Bulk seawater and sea surface microlayer sampling

Seawater samples were taken from different environments, i.e., ice-free ocean, marginal ice zone (MIZ), within the ice pack or from meltponds. In case of the former three the samples were taken a few hundred meters away from the position of the Polarstern using a Zodiac boat, while the meltponds on the ice floe could be reached on foot. BSW samples were typically

taken from a depth of one meter with the help of a sealable bottle on a telescopic rod, with the exception of shallow meltponds, where the sample were taken near the ground (for details see Zeppenfeld et al., 2019). SML samples were collected with a





glass plate sampler (Zeppenfeld et al., 2019; Van Pinxteren et al., 2017). The glass plate is dipped into the water body, slowly withdrawn and the surface film, which clings to the sides of the glass plate, is wiped off the plate into a sample container with a Teflon® wiper.

The seawater sampling was conducted on a daily basis. The SML and bulk seawater samples were taken at the same time and location with the only exceptions of shallow meltponds where no samples from one meter depth could be taken as well as days with harsh weather when no surface film could form. 42 SML samples and 42 bulk seawater samples were collected during the campaign. A further description of the seawater sampling, and a chemical and microbiological analysis of the samples can be found in Zeppenfeld et al. (2019). A list of the seawater samples can be found in Tab. S2 in the SI.

### 2.2.3   Fog sampling

Fog was collected with the Caltech Active Strand Cloud Collector Version 2 (CASCC2; described in Demoz et al., 1996). The CASCC2 is a non-selective sampler that catches hydrometeors by impaction on Teflon®strands (508 $\mu$m diameter). Droplets caught on the strands are gravitationally channeled into a Nalgene bottle. The instrument operates with a flow rate of approximately 5.3 m$^3$ min$^{-1}$ resulting in a 50% lower cut-off size of approximately 3.5 $\mu$m. During daytime on leg 1 the sampler turned on every time the visibility decreased significantly and was running continuously during the night. On leg 2 the sampler was running continuously and the sample bottle was changed whenever a significant amount of sample material was collected and the fog event was over. In all cases the sampler was rinsed with ultrapure water after a fog event was sampled and the sample bottle changed. During the entire campaign, 22 samples were collected, about two thirds of them on the second leg alone. A list of all fog samples can be found in Tab. S3 in the SI.

### 2.3   INP analysis

#### 2.3.1   Sample preparation

Samples stored at -24°C were thawed only to perform the measurements. The measurements were performed on the same day as the thawing, and the remaining sample material was refrozen at the end of the day on which the measurements were completed.

The polycarbonate filters are put in a centrifuge tube along with 3 mL of ultrapure water and are shaken in an oscillating shaker for 15 minutes in order to extract the particles from the filter and bring them into suspension. Then 100 $\mu$L of that suspension are removed for the analysis with the Leipzig Ice Nucleation Array (LINA; described section 2.3.3). For the analysis with the Ice Nucleation Droplet Array (INDA; described in the section 2.3.4), the remaining 2.9 $\mu$L of the suspension are made up to a total of 6 mL with ultrapure water and shaken again as before. The reason for this procedure is to use as little water as viable, i.e., to dilute the sample as little as possible.

Sea and fog water samples do not require any preparation and can be directly measured with either setup.



### 2.3.2 Test for heat-labile INPs

After the initial measurement arbitrarily selected samples were chosen to test for the presence of heat-labile INPs in the samples. The sample solution was sealed in an centrifuge tube and placed in an oven. The sample was heated at 95°C for 1 h
and subsequently analysed with the LINA device (described section 2.3.3).

### 2.3.3 Leipzig Ice Nucleation Array (LINA)

LINA is a Droplet Freezing Assay (DFA), the design of which is based on a DFA called BINARY by Budke and Koop (2015). An array of 90 droplets with a typical volume of 1 $\mu$L of the sample suspension is placed onto a hydrophobic glass slide (40 mm diameter). Each droplet is within its individual compartment made from a perforated, anodized aluminum plate and covered
with another glass slide. In this way it can be ensured that droplets do not interact during the freezing process, e.g., via ice seeding by frost splintering or the Bergeron-Wegener-Findeisen process. Furthermore, droplet evaporation is minimized. At a cooling rate of 1 °C min$^{-1}$ the sample droplets are cooled by a 40 x 40 mm$^2$ Peltier element inside a freezing stage (LTS120, Linkam Scientific Instruments, Waterfield, UK). The freezing stage is coupled with a cryogenic water circulator (F25-HL, Julabo, Seelbach, Germany) in order to achieve temperatures below -25°C down to the temperature at which homogeneous
freezing occurs naturally, i.e. -38°C. A thin layer of squalene oil thermally connects the Peltier element and the glass slide with the droplets on top. The freezing stage itself consists of a gas tight aluminium housing, which is purged with dry, particle-free air during the measurement. A LED dome lighting (SDL-10-WT, MBJ-Imaging GmbH, Hamburg, Germany) is used for shadow-free illumination of the droplets. A charge-coupled device camera is mounted at the apex of the dome and takes images every 6 s which corresponds to a temperature resolution of 0.1°C if cooled with 1 °C min$^{-1}$. An aperture below the dome blocks
the light partially and creates a ring-shaped reflection in each droplet. This is used as detectable feature that vanishes upon freezing of the droplet. A custom Python algorithm then evaluates each image in terms of the number of frozen droplets, $N_f$, in each individual image. As every image corresponds to a certain temperature the frozen fraction at the respective temperature, $f_{ice}(T)$, can be easily derived.

### 2.3.4 Ice Nucleation Droplet Array (INDA)

The basic design of the INDA device is inspired by Conen et al. (2012), but as suggested in Hill et al. (2016), Polymerase Chain Reaction (PCR) plates instead of individual tubes were used. In each of the 96 wells of the PCR plate 50 $\mu$L sample material is filled. Then the PCR plate is sealed with a transparent cover foil and immersed in the bath of a cryostat (FP45-HL, Julabo, Seelbach, Germany) in a way that the wells itself are surrounded by refrigerant (ethanol), but not so deep that the PCR plate would be completely submerged. The PCR plate is illuminated from below which makes the phase change of the sample
suspension visible as darkening of the respective well. The temperature of the refrigerant lowered with a rate of ca. 1 °C min$^{-1}$, while simultaneously the temperature is recorded and a on-top mounted camera takes pictures at 0.1 °C intervals. The images are then again evaluated with a custom Python algorithm for $N_f$ in order to derive $f_{ice}(T)$.





### 2.3.5 INP number concentrations $N_{\mathrm{INP}}$

Cumulative number concentrations of INPs per volume of sample as a function of temperature were calculated for each exper-
iment utilizing the equation given in Vali (1971):

$$N_{\mathrm{INP}}(T) = \frac{-\ln(1 - f_{\mathrm{ice}})}{V_{\mathrm{drop}}} \tag{1}$$

with $f_{\mathrm{ice}} = \frac{N_{\mathrm{frozen}}(T)}{N_{\mathrm{total}}}$, where $N_{\mathrm{total}}$ is the number of droplets, and $N_{\mathrm{frozen}}$(T) the number of frozen droplets at temperature $T$.
With the given number of droplets ($N_{\mathrm{total}} = 90$) and volume ($V_{\mathrm{drop}}$=1 $\mu$L), the upper and lower limits of the detectable range
of LINA are $1.12 \cdot 10^4$ and $4.5 \cdot 10^6$ L$^{-1}$(water), whereas $2.1 \cdot 10^2$ and $9.1 \cdot 10^4$ L$^{-1}$(water) are the limits for INDA ($N_{\mathrm{total}} = 96$; $V_{\mathrm{drop}}$
$= 50\,\mu$L). The temperature values of the seawater samples were corrected for freezing point depression due to the salt content
as described in Koop and Zobrist (2009).

In case of the atmospheric filter samples in order to derive atmospheric $N_{\mathrm{INP}}$, the denominator in Eq.1 needs to be modified so
that it represents the volume of air distributed into each droplet:

$$N_{\mathrm{INP}}(T) = \frac{-\ln(1 - f_{\mathrm{ice}})}{\frac{V_{\mathrm{air}}}{V_{\mathrm{wash}}} * V_{\mathrm{drop}}} \tag{2}$$

where $V_{\mathrm{air}}$ is the air volume sampled onto one filter and $V_{\mathrm{wash}}$ is the volume of water the particles were rinsed off with and
suspended in.

### 2.4 Collocated Measurements and Supporting Observations

In addition to the sampling of INPs, the physico-chemical properties of the prevailing atmospheric aerosol particles were
measured inside a temperature controlled measurement container located on the monkey island of the RV Polarstern. The tem-
perature inside the container was held at ca. 24°C, while the aerosol inlet was heated to 30°C to prevent icing. The aerosol inlet
consists of a 6 m long stainless steel tubing (inner diameter of 40 mm), which faces upwards at a 45°angle to the bow of the
ship. The flow through the inlet was set to 40 L min$^{-1}$ (Reynolds number < 2000). With an isokinetic splitter the aerosol was
distributed between the different instruments. The aerosol instrumentation relevant to this study included: a mobility particle
size spectrometer (MPSS) to measure particle number size distributions (PNSD), a condensation particle counter (CPC) to
measure total particle concentration ($N_{\mathrm{tot}}$), and a cloud condensation nuclei counter (CCNC) to measure the concentrations of
cloud condensation nuclei ($N_{\mathrm{CCN}}$).

PNSDs in the size range between 10 and 800 nm were measured with a TROPOS-type MPSS (Wiedensohler et al., 2012).
The time resolution of an up- and down-scan was 5 min. PNSDs were derived with the inversion algorithm by Pfeifer et al.
(2014) and corrected for transmission losses as well as counting efficiencies according to Wiedensohler et al. (1997). The sizing
of the MPSS was calibrated according to Wiedensohler et al. (2018) at regular time intervals during the campaign (for further
details on the MPSS and the measurement container refer to Kecorius et al., 2019). $N_{\mathrm{tot}}$ was measured with a CPC (Model
3010, TSI Inc., Shoreview, USA; lower cutoff: 10 nm). A CCNC (CCN-100, DMT, Boulder USA Roberts and Nenes, 2005)





was used to measure $N_{\text{CCN}}$ at six different supersaturations (SS; 0.1%, 0.15%, 0.2%, 0.3%, 0.5%, 1%,). Each SS was sampled

for 10 min and averaged over that period, hence a certain SS has an time resolution of 1 hour. The instrument was calibrated

with ammonium sulfate particles before and after the campaign according to the ACTRIS protocol (Gysel and Stratmann, 2013).

In addition to the off-line INPs analysis of the filter samples, also the SPectrometer for Ice Nuclei (SPIN; Droplet Mea-

surements Techniques, Boulder, CO, USA) was deployed to measure $N_{\text{INP}}$ on-line (Garimella et al., 2016). SPIN was placed

within a measurement container. Together with the other aerosol instrumentation was the aerosol was fed to SPIN through one

main inlet, but with additional subsequent drying of the aerosol. SPIN sampled in half-hourly intervals of constant temperature

and relative humidity and each sampling condition was repeated three times within 24 h. The SPIN dataset is also part of the

overview of global ship-borne INP measurements by Welti et al. (2020).


Na$^+$ and Cl$^-$ mass concentration on size-resolved ambient aerosol particles were measued from five-stage Berner impactor

samples (mounted outside of the measurement container, setup described in detail in Kecorius et al. (2019)) by ion chromatog-

raphy (ICS3000, Dionex, Sunnyvale, CA,USA), as described in Müller et al. (2010) in more detail. Ion chromatography was

also used to determine the salinity of the seawater samples.

5-day air mass back trajectories were calculated using the Hybrid Single-Particle Lagrangian Integrated Trajectory (HYSPLIT)

model (Rolph et al., 2017; Stein et al., 2015). As input for the model, the GDAS1 meteorological fields (Global Data Assimi-

lation System; 1°latitude/longitude; 3-hourly) were used. Trajectories were initiated at 50 m, 250 m and 1000 m every hour.

The sea ice concentration at the position of the Polarstern was determined with the sea ice concentration product of the EU-

METSAT Ocean and Sea Ice Satellite Application Facility (OSI-401-b: SSMIS Sea Ice Concentration Maps on 10 km Polar

Stereographic Grid Tonboe et al., 2017)

Chlorophyll-a concentration were derived from the vessel's Ferrybox system and also from satellite remote sensing (Aqua

MODIS, NPP, L3SMI, Global, 4km, Science Quality, 2003-present, 8 Day Composite (July 8 to July 16)).

## 3 Results and Discussion

### 3.1 Atmospheric INP concentrations $N_{\text{INP}}$

A time series of atmospheric $N_{\text{INP}}$ at selected temperatures derived from filter samples and on-line measurements with SPIN

is shown in Fig. 2. The light gray areas, mark the periods where the Polarstern was beyond the ice edge within the ice pack

and the dark gray area shows the time of the ice floe camp. Overall $N_{\text{INP}}$ is the highest at the beginning of the campaign, in

between both legs at the harbour of Longyearbyen and upon entering the ice-free ocean again towards the end of the second

leg. The lowest concentrations occurred when the vessel was within the ice pack. It can also be seen that at a given time, peaks

appear or disappear depending on temperature, indicating that different populations of INPs contribute at warmer or colder

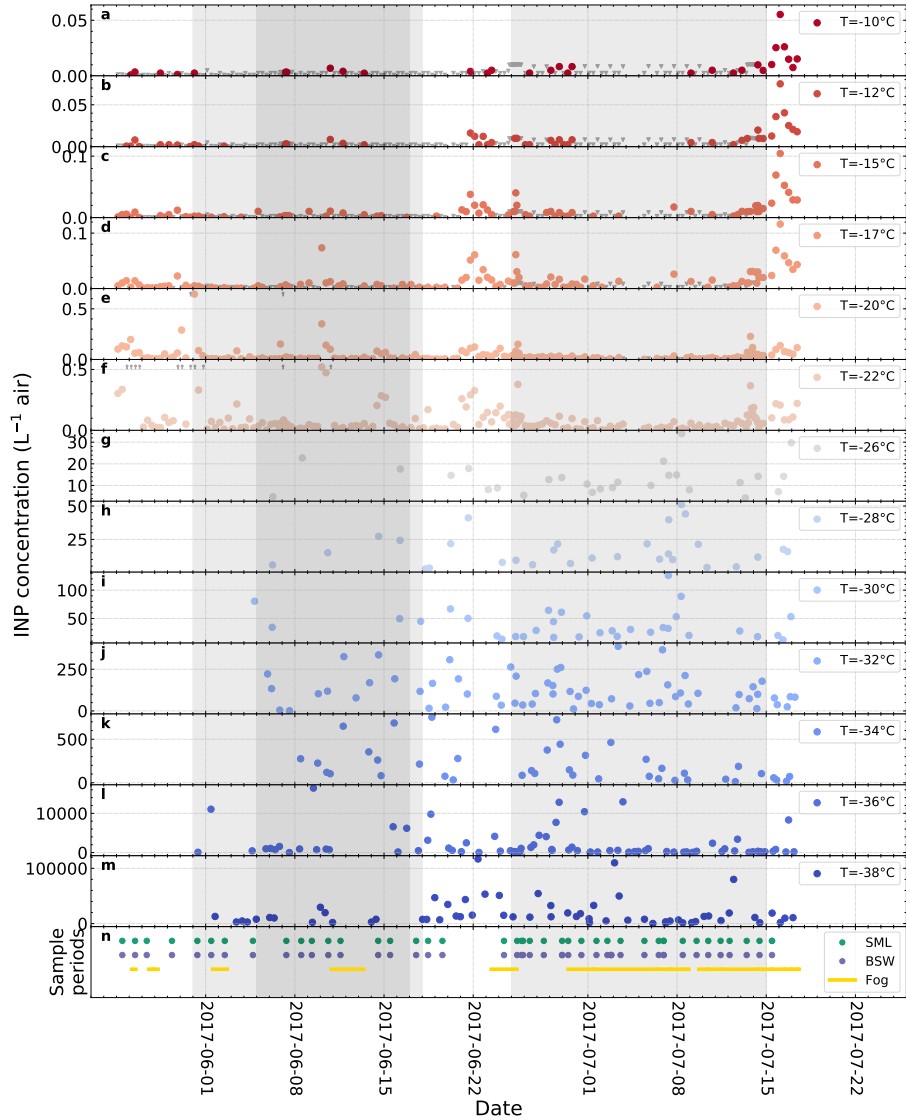

**Figure 2.** Time series of atmospheric $N_{INP}$ at different T derived from filter samples (-22°C and above) and SPIN measurements (-26°C and below). The bottom panel contains markers for the sample collection times of the SML, BSW and fog water samples. The light gray areas, mark the periods where the Polarstern was beyond the ice edge within the ice pack and the dark gray area shows the time of the ice floe camp.

temperatures.

Fig. 3 shows the $N_{INP}$ freezing spectra for the atmospheric filter samples measured with LINA (circle markers), as well as $N_{INP}(T)$ measured with SPIN (cross markers). The color represents the environment in which the sampling took place, based 260 on the sea ice concentrations at the location of the Polarstern (yellow = ice-free ocean; blue = ice pack; purple = marginal





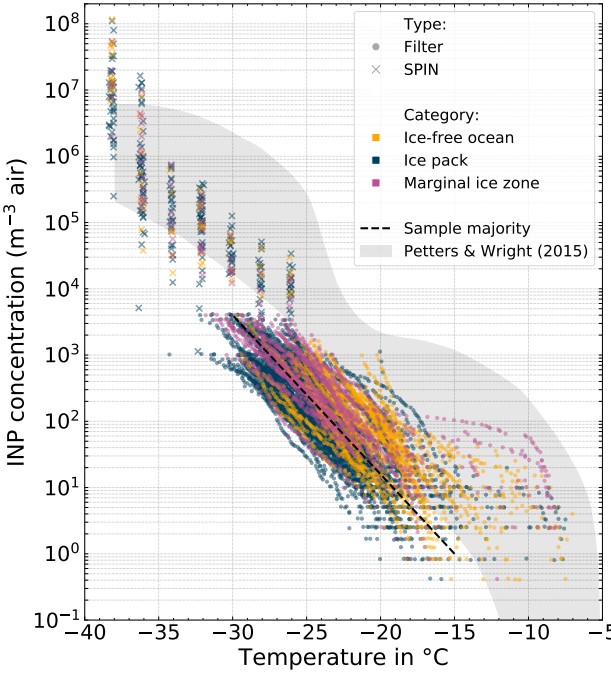

**Figure 3.** All cumulative INP spectra derived from atmospheric filter samples measured with LINA (circle marker), as well as $N_{INP}$(T) measured with SPIN (cross marker). The color code refers to the environment the sample was taken from (yellow = ice-free ocean; blue = ice pack; purple = marginal ice zone). The majority of the filter samples are clustered around a line which is shown as black dashed line. The range of $N_{INP}$ for mid-latitudes by Petters and Wright (2015) is shown as gray shaded are for reference.

ice zone, MIZ). The MIZ is defined as the transitional zone between open sea and dense drift ice. It spans from 15% to 80% of the sea surface being covered with ice. The area north of the MIZ is classified as the ice pack and the area south of the MIZ as ice-free ocean. As the filter samples were collected over the course of several hours on a often moving vessel, the sample environment might change during sampling. In such cases the sample was labeled according to the environment which

accounts for most of the sampling time. The range of $N_{INP}$ for mid-latitudes by Petters and Wright (2015) is shown as gray shaded are for reference.

At any particular temperature $N_{INP}$ varies between two to three orders of magnitudes. The variability tends to be higher at warmer temperatures compared to colder temperatures: At -10°C $N_{INP}$ varies between $4*10^{-1}$ and $6*10^1$ m$^{-3}$, at -17°C between $4*10^{-1}$ and $1*10^2$ m$^{-3}$, and at -25°C between $3*10^1$ and $2*10^3$ m$^{-3}$. It can be seen that the majority of the samples

are clustered around a line ranging roughly from (-15°C; 1 m$^{-3}$) to (-30°C; $4*10^3$ m$^{-3}$). But also highly ice active filter samples featuring $N_{INP}$ as high as $6*10^1$ m$^{-3}$ at -10°C were observed. These tend to be associated more often with the MIZ (purple symbols) and the ice free ocean (yellow symbols) environment than with the ice pack. We will describe these highly ice active samples in more detail in section 4. In comparison to the range of $N_{INP}$ from mid-latitudes by Petters and Wright (2015), the filter derived $N_{INP}$ are lower for temperatures below ca. -20°C, but similar at warmer temperatures. For a given temperature



$N_{\mathrm{INP}}$ measured with SPIN falls within the lower half of the $N_{\mathrm{INP}}$ range by Petters and Wright (2015), with the exception of the two lowest temperature steps.

In Fig. 3 it can be seen that some LINA freeezing spectra go up to higher values ($4 * 10^3 \, \mathrm{m^{-3}}$) than others (ending at $1 * 10^3 \, \mathrm{m^{-3}}$). The cause of this lies in the measurement principle itself: DFAs only measure $N_{\mathrm{INP}}$ per volume of water in a certain concentration range determined by the specific setup configuration. With the known volume of air sampled onto one filter, these
concentrations per volume of water are then scaled to atmospheric concentrations per volume of air. Hence differing ranges of resulting values are caused by systematic differences in $V_{\mathrm{air}}$. In our case we collected samples for 8 h or 2 h and since the flow rate is relatively constant, $V_{\mathrm{air}}$ of the 8 h samples is about 4 times larger than of the 2 h samples, which causes also the different reported ranges in atmospheric $N_{\mathrm{INP}}$ as seen in Fig. 3. It should also be mentioned that the upper and lower ends of the freezing spectra shown in this work only represent the limits of our detectable range and do not imply that outside these
limits no higher $N_{\mathrm{INP}}$ or lower $N_{\mathrm{INP}}$ existed.

The test for heat-labile INPs (Fig. S6 in the SI) demonstrates that ice activity of the samples is reduced when heated for 1 h at 95°C. Especially INPs that nucleated ice at temperatures above -15°C are gone after the heating. This is an indicator for the presence of biogenic, proteinaceous INPs in that temperature range.

### 3.1.1  INPs at low temperatures

In the previous section we described that at warmer temperatures like -10°C, the most ice active samples are more often associated with the MIZ and the ice-free ocean rather than the ice pack as an environmental setting. In comparison, at the lower temperatures measured with SPIN (cross markers in Fig. 4) no correlation with the environmental setting is found. However, in global context the level of $N_{\mathrm{INP}}$ at these low temperatures is remarkable by itself as shown in Fig. 4. That figure shows $N_{\mathrm{INP}}$ in the Arctic at -32°C measured with SPIN during PS106, but also SPIN data by Welti et al. (2020) of a transect from Bremerhaven
(Germany) to Cape Town (South Africa) along the western coast of Africa. It is striking that at these low temperatures $N_{\mathrm{INP}}$ in the Arctic are in the same order of magnitude as in the outflow region of mineral dust from the Saharan desert. We see that as indication that mineral dust and hence terrestrial sources greatly contribute to $N_{\mathrm{INP}}$ at low temperatures.

### 3.2  INPs in sea surface microlayer and bulk seawater

Fig. 5 shows $N_{\mathrm{INP}}$ per volume of water in SML and BSW. Again, the color code refers to the environment the sample was
taken from (yellow = ice-free ocean; blue = ice pack; purple = MIZ; green = melt pont). The gray box indicates the range of values reported in an earlier study by Wilson et al. (2015) in the Arctic, albeit they did not separate their samples into different environments.

Both, SML and BSW show a high intersample variability. Concentrations for both vary at least between 2 and 3 orders of magnitude at any temperature. Some samples initiate freezing clearly above -10°C (highest observed freezing onset was at -
5.5°C), while for other samples, freezing starts only at temperatures below -15°C. It is worth mentioning that the concentration range of INP reported by Wilson et al. (2015) is not directly comparable to the measurements we present, because due to a different measurement setup they have different limits of their detectable range. Nevertheless, it can be seen that their SML





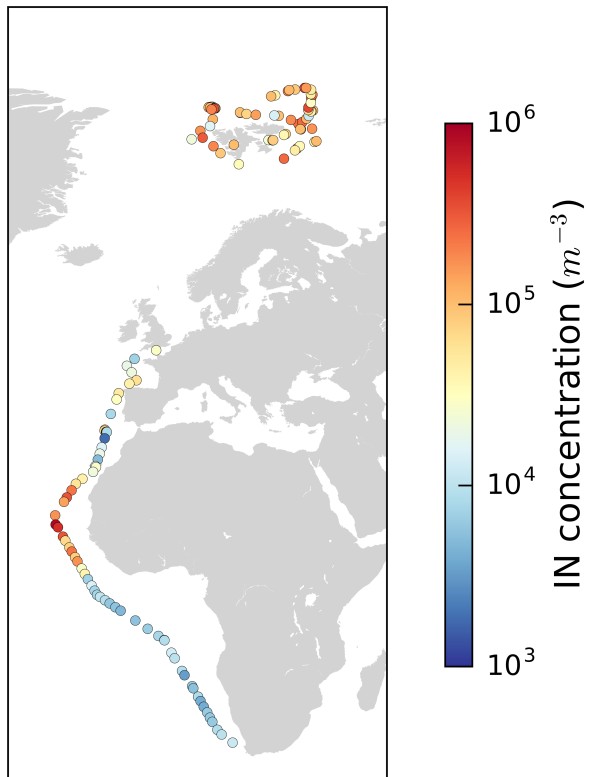

**Figure 4.** Map with color-coded $N_{\mathrm{INP}}$ in the Arctic at -32°C measured with SPIN during PS106 and also SPIN data of a transect from Bremerhaven (Germany) to Cape Town (South Africa) along the western coast of Africa (Welti et al., 2020).

samples contain up to two orders of magnitude higher concentrations of INPs ice active at high temperatures (above ca. -10°C). Interestingly, some of our samples stand out, i.e., feature significantly higher ice activities than the others (see Fig. 5). For the

SML two groups of samples (encircled in Fig. 5) and a single sample (between cluster 1 and 2), and for the BSW one group of two samples (labeled as cluster 3), can be identified as different from the majority. It is noticeable that SML samples from the known biologically active MIZ, belong mostly to the group of samples that stand out from the rest. Also the overall most active SML sample originates from the MIZ and its connection to the corresponding atmospheric filter samples is discussed in more detail in the case study in section 4. The high variability of the INP concentrations in SML and BSW in our view is a clear hint

towards the sporadic occurrence of INPs in these compartments.

The SML has been found to be enriched in particulate organic matter and surface-active substances compared to the underlying bulk seawater, with enrichment factors (EF) of up to 10 and 50 respectively being reported (Engel et al., 2017; Kuznetsova and Lee, 2002). And, as described in the introduction, the SML is known to be highly ice active. It is therefore an interesting question whether INPs are also enriched in the SML compared to BSW, and whether enrichment is a general feature in all





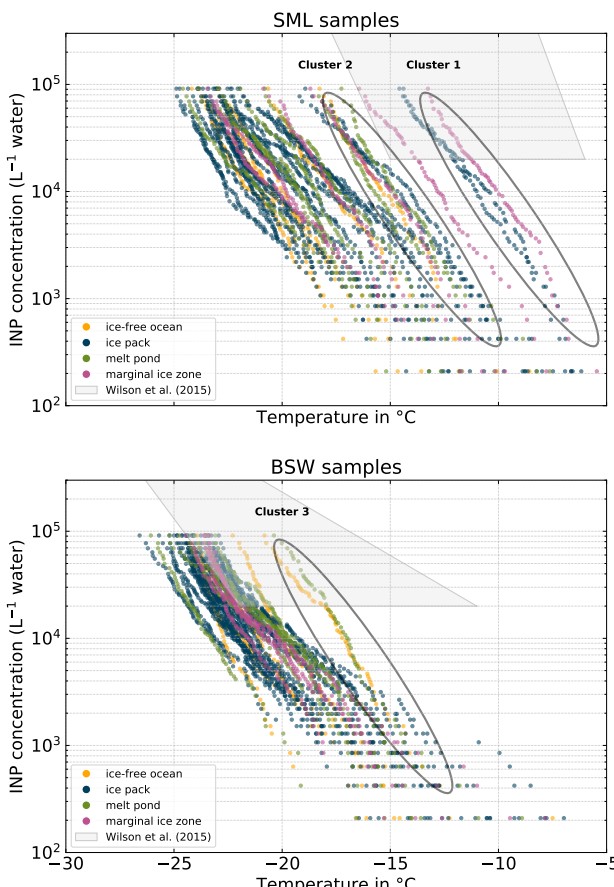

**Figure 5.** $N_{INP}$ in SML and BSW measured with the INDA device. Samples are categorized according to the environment (ice-free ocean, ice pack, melt pond, marginal ice zone) the samples was taken from. The gray box indicates the range of values reported by Wilson et al. (2015) for the Arctic

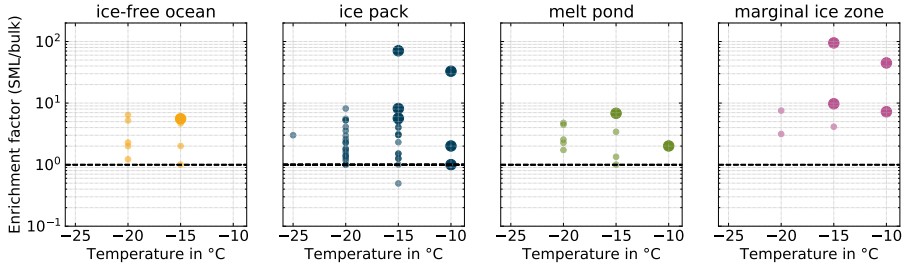

**Figure 6.** Enrichment factors for all pairs of SML and BSW samples divided into separate panels by their environmental setting. Larger markers correspond to the pairs of samples for which either the SML or the BSW sample stands out in terms of ice activity.





samples or whether it is restricted to certain situations. To answer this question we derived INP enrichment factors for the

SML, based on corresponding SML and BSW samples as:

$$EF_{\mathrm{INP}}(T) = \frac{N_{\mathrm{INP,SML}}(T)}{N_{\mathrm{INP,BSW}}(T)} \qquad (3)$$

Fig. 6 depicts the calculated EFs at selected temperatures. We observe that the majority of SML samples are enriched in INPs

compared to the underlying BSW. The majority of EFs falls into the range between 1 and 10, with only four occurrences of

higher values. The highest EF we found was 94.97. There are seven occurrences of $EF = 1$ and only one of $EF \leq 1$. This

result is similar to Wilson et al. (2015), who only observed enrichment and no depletion of INPs in the SML. On the other hand

the study by Irish et al. (2017) also reports few cases of INPs depletion in the SML. But, as Gong et al. (2020) pointed out,

direct comparisons of EFs between different studies are are difficult since methodological differences might be of importance.

The larger markers in Fig. 6 indicate samples where the SML showed significantly higher ice activity compared to the others,

i.e., higher INP concentrations (see above). Interestingly, almost exclusively the highly ice active SML samples are the samples

which feature the highest EFs, suggesting that enrichment could be an important factor in controlling SML ice activity.

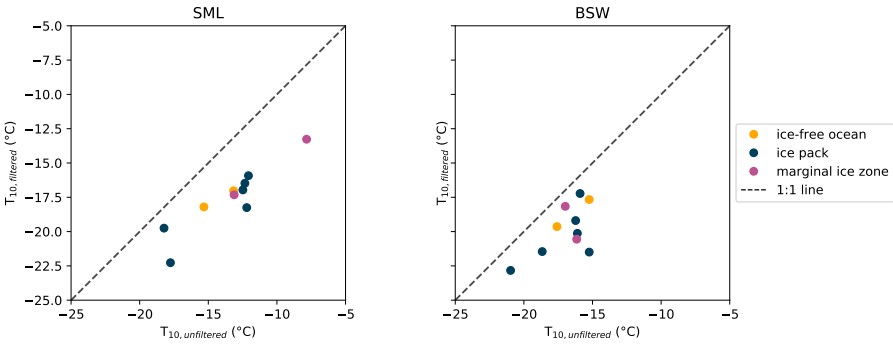

**Figure 7.** Comparison of filtered and unfiltered SML and BSW samples. Shown are the $T_{10}$ values for corresponding samples. Symbols below the 1:1 line indicate that the filtered sample is less ice active.

Filtrations of ten randomly selected SML and twelve BSW samples were created and analysed for $N_{\mathrm{INP}}$ to find indications

concerning the size of the INPs present in the samples. The samples were filtered with $0.2\,\mu$m PTFE syringe filters (Puradisc

25, Whatman). While the individual sample was chosen randomly it was ensured that all sample environments (ice-free ocean,

ice pack, melt pond, marginal ice zone) were considered. Fig. 7 shows a scatter plot of the $T_{10}$ values, i.e., the temperature

where 10% of the droplets are frozen, of the filtered and unfiltered samples. If a sample falls below the 1:1 line it indicates that

the filtration reduced the ice activity and the distance to the 1:1 line in x-direction is a measure of how strong the reduction

in ice activity is. The complementary plots of the $T_{50}$ and $T_{90}$ can be found in the SI. Throughout all samples a reduction of

the freezing temperatures can be seen. Also, the more ice active the unfiltered sample was, the larger the shift towards lower

temperatures tends to be (see also Fig. S4 in the SI). The most ice active sample shifted by around 5°C, while those with lower





initial ice activity only are decreased by approximately 2°C. This clearly indicates that a high fraction of the INPs are larger or at least associated with particles larger than $0.2\,\mu$m.

## 3.3 INPs in fog water

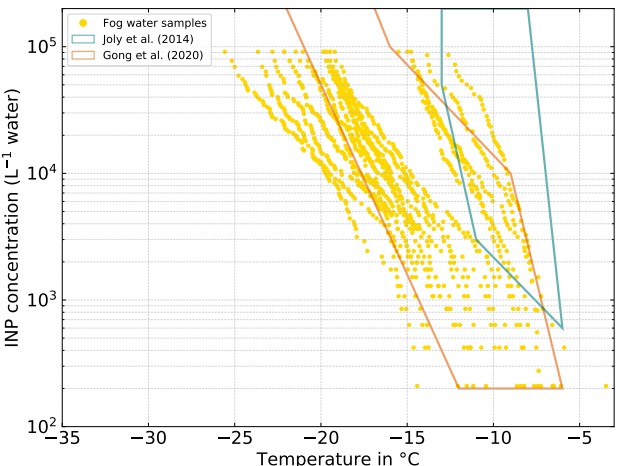

**Figure 8.** $N_{\mathrm{INP}}$ in fog water samples. The teal and orange polygons show the range of values observed by Joly et al. (2014) and Gong et al. (2020) respectively.

Analogous to the SML and BSW samples, $N_{\mathrm{INP}}$ was also determined in collected fog water samples. At -10°C we find $N_{\mathrm{INP}}$ between the lower limit of our detectable range of $2*10^2$ and $2*10^4\,L^{-1}$. At -15°C $N_{\mathrm{INP}}$ between $6*10^2\,L^{-1}$ and the upper limit of our detectable range $9*10^4\,L^{-1}$, were observed. At -20°C values between $1*10^4\,L^{-1}$ and the upper limit of our detectable range, $9*10^4\,L^{-1}$ were found. 14 fog samples (63.6% of all fog samples) have a freezing onset above -10°C suggesting the presence of biogenic INPs as mineral dust only starts to contribute to the INPs population at temperatures below -15°C (e.g.

Murray et al., 2012; O'Sullivan et al., 2018). The highest freezing onset we observed in a sample was at -3.47°C. The samples are divided into two groups by a clearly recognizable gap. The occurrence of these two groups could not directly be related to meteorological parameters. However, as will be discussed in section 3.3.1, the group of more ice active fog samples may be associated with the more ice active atmospheric filter samples.

In general the fog samples tend to be more ice active and show higher $N_{\mathrm{INP}}$ at a given temperature than the seawater samples

presented in section 3.2. A qualitatively similar observation was already made by Schnell (1977). For seawater samples they collected near Nova Scotia (Canada) they found that some of the samples were very ice active, although the majority of their seawater samples contained no INP active at temperatures warmer than -14°C. On the other hand half of their fog water samples were ice active at temperatures above -10°C with the most ice active sample initiating freezing at -2°C. Schnell (1977) also described that they found $N_{\mathrm{INP}}$ in seawater, fog and air to vary independently from each other. An observation that also largely

applies to this study, but a more detailed investigation of the relation between $N_{\mathrm{INP}}$ in the different compartments is presented





in the following sections (3.3.1 and 3.4).

The $N_{\text{INP}}(T)$ we observed in Arctic fog water is similar to what Gong et al. (2020) found in cloud water samples on the Cape Verde Islands, but tends to be lower than what was observed by Joly et al. (2014), who measured at Puy de Dôme (France) and reported a correlation between high concentrations of biological particles and INP concentrations. However the freezing onset

temperature of around -6°C is almost identical in the three studies.

### 3.3.1  Connecting INPs in clear or fog-free air to fog samples

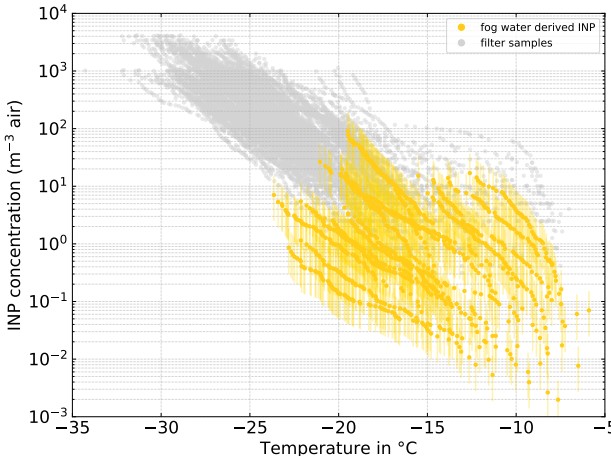

**Figure 9.** Fog water derived $N_{\text{INP}}$ in air. $N_{\text{INP}}$ was derived from Eq. 4 and 5 with the median $N_{\text{CCN}}(\text{SS}=0.15\%)$ during the time of each fog sample and an average droplet diameter ($\text{d}_{\text{drop}}$) of 17 $\mu$m. The error bars show the range with $\text{d}_{\text{drop}}$ of 12 $\mu$m and 22 $\mu$m respectively. See section 3.3.1 for details on the derivation method.

In this section we relate and compare $N_{\text{INP}}$ in fog water samples with those measured in fog-free air (see section 3.1), following the procedure introduced in Gong et al. (2020), which is briefly described in the following. The number concentration of CCN ($N_{\text{CCN}}$) at a particular supersaturation (SS) is used as proxy for the fog droplet number concentration. Furthermore,

Gong et al. (2020) made the legitimate assumption, that all INPs act as CCN. Together with an estimated fog droplet diameter ($\text{d}_{\text{drop}}$), the volume of fog water per volume dry air, $\text{LWC}_{\text{fog}}$ can be calculated as follows:

$$LWC_{\text{fog}} = N_{\text{CCN}} * \pi/6 * d_{drop}^3 \tag{4}$$

For determining $N_{\text{CCN}}$ a SS needs to be defined. Since fog, unlike clouds, is characterized by low updrafts, SS is also typically low (0.02% - 0.2% Pruppacher and Klett, 2010). Thus we choose $N_{\text{CCN}}$ measured at SS = 0.15% as proxy for the droplet

number concentration. Please note that we do not use $N_{\text{CCN}}$ measured at SS = 0.1%, because after the removal of data points due to quality assurance, the data coverage for SS = 0.15% is significantly better than for SS = 0.1%.

Remote sensing studies of Arctic cloud droplets sizes report typical diameters between 14 and 20 $\mu$m (Bierwirth et al., 2013; Shupe et al., 2001; King et al., 2004) and in-situ observations found values between 12 and 22 $\mu$m. Hence we use 17 $\mu$m as an





average $d_{drop}$ and vary between 12 and 22 $\mu$m. With that we calculate a range of $LWC_{fog}$, which then is further used to derive

the INP number concentration in air, $N_{INP,air}$, based on the INP concentration in fog water, $N_{INP,fogwater}$:

$$N_{INP,air} = LWC_{fog} * N_{INP,fogwater} \tag{5}$$

Fig. 9 depicts $N_{INP}$ as determined from the clear air filter samples with gray symbols and the ones derived from the fog water samples in yellow. Overall, measured and derived $N_{INP}$ are in good agreement. Unfortunately, as multiple atmospheric filter

samples were taken during the collection time of a single fog sample, an unambiguous attribution of a filter to a fog sample is difficult. Therefore, here we can only report the half-quantitative observation that the freezing spectra of the atmospheric filter samples taken concurrently with the most ice active fog samples and the spectra derived from the fog water samples feature similar shapes, with the shapes themselves and the onset of freezing at temperatures above -10°C suggesting the presence of biogenic INPs. This clearly points at the same or at least similar, partly biogenic INP populations being present in both, fog

droplets and atmospheric aerosol particles. Also Gong et al. (2020) found general agreement between $N_{INP}$ in the air and $N_{INP}$ derived from, in their case, cloud water samples. They further observed that highly ice active particles are activated into cloud droplets during cloud events and then can be found in the cloud water. It is likely that a similar process occurs during our fog events.

It should be noted that if $N_{CCN}$ changes significantly during the sampling time of the respective fog sample, the fog-derived

atmospheric $N_{INP}$ is directly affected. Such an instance can be seen in the lowest fog-derived INP spectra in Fig. 9, where the low average $N_{CCN}$ lead to a deviation of around one order of magnitude in comparison to the atmospheric sample.

### 3.4   Connecting atmospheric INPs to sea spray

In order to assess the ocean as possible source of atmospheric INPs, we derive potential atmospheric $N_{INP}$ by virtually dispersing the characterized seawater samples as sea spray (Irish et al., 2019; Gong et al., 2020). This thought experiment can be

paraphrased as follows: If the seawater samples including all their INPs would be directly dispersed into the air, scaled by the measured relation between salt in the air and in the water, what would be the resulting $N_{INP}$ in the air?

For this approach we use the amount of NaCl present in the atmospheric aerosol particles (derived from chemical analysis of Berner impactor samples) in relation to the amount of NaCl present in the seawater as a scaling factor to translate $N_{INP}^{seawater}$ into atmospheric $N_{INP}$. In this simple model, no enrichment of INP is accounted for in the course of sea spray production. The

sea spray derived INP concentrations ($N_{INP}^{sea\,spray,\,air}$) are calculated as:

$$N_{INP}^{seaspray,air} = \frac{NaCl_{mass,air}}{NaCl_{mass,seawater}} * N_{INP}^{seawater}, \tag{6}$$

where $NaCl_{mass,\,air}$, and and $NaCl_{seawater}$ are the mass concentrations of sodium chloride in corresponding air and seawater samples, respectively. $NaCl_{mass,\,air}$ varied between 0.04 and 1.9 $\mu$g m$^{-3}$ during the campaign with an average of 0.48 $\mu$g m$^{-3}$. The average $NaCl_{seawater}$ of all SML and BSW samples is 32.5 g L$^{-1}$ with actual concentrations varying between 25.7 and

34.5 g L$^{-1}$. $NaCl_{seawater}$ was derived from the salinity of the samples with the simplifying assumption that NaCl is the only salt





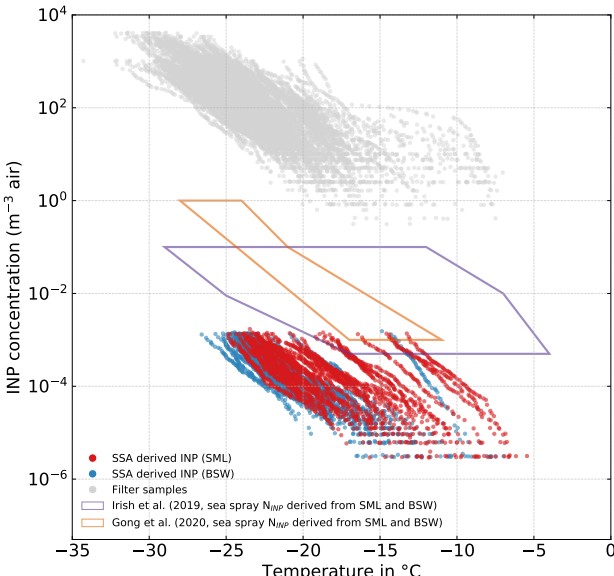

**Figure 10.** Sea spray derived atmospheric $N_{INP}$ (red symbols = derived from SML samples; blue symbols = derived from BSW samples). The gray symbols show the filter derived atmospheric $N_{INP}$. The orange and purple polygon indicates the range of SSA derived $N_{INP}$ by Irish et al. (2019) and Gong et al. (2020) respectively. Samples from meltponds are excluded.

in the sea water. This assumption is justified as non-NaCl salts represent only minor constituents of the sea water. Samples from meltponds are excluded here and also in the following as they are mostly fresh water and therefore not suited for this approach that is based on NaCl concentration.

Fig. 10 shows atmospheric, filter derived $N_{INP}$ in gray and the sea spray derived $N_{INP}^{\text{sea spray, air}}$ (red symbols correspond to
SML samples and blue ones to BSW samples). As can be seen, $N_{INP}^{\text{sea spray, air}}$ falls mostly in the range between $10^{-6}$ and $10^{-2}$ m$^{-3}$, which is approximately 4 to 5 orders of magnitude lower than the atmospheric $N_{INP}$ derived from our atmospheric filter samples. Our concentration range of $10^{-6}$ to $10^{-2}$ m$^{-3}$ is also roughly two orders of magnitude lower than the ranges reported by Gong et al. (2020), who sampled near the subtropical islands of Cape Verde during late summer, and Irish et al. (2019), who measured in the Canadian Arctic during early summer. These differences could be due to the geographical settings
of the samples being vastly different, even for the Arctic measurements by Irish et al. (2019). Irish et al. (2019) took samples comparatively close to the shore mainly in the Nares Strait and Baffin Bay during Summer with no extensive sea ice cover present, whereas we sampled mostly within the ice pack hundreds of nautical miles away from bigger lands masses. But even if the ranges given in Gong et al. (2020) and Irish et al. (2019) are considered, our atmospheric filter-derived $N_{INP}$ are still orders of magnitude higher than any sea spray derived $N_{INP}$. This indicates that sea spray aerosol as sole source is not sufficient
to explain atmospheric $N_{INP}$ without significant enrichment of INPs during sea spray production. To the authors' knowledge, there are no studies available on the enrichment of INPs in sea spray aerosol (SSA). However, studies about the enrichment of organic matter exist. Enrichment factors of $10^4$ to $10^5$ (in relation to mass) are reported for submicron SSA (Keene et al.,




2007; Van Pinxteren et al., 2017) and $10^2$ for supermicron SSA (Quinn et al., 2015; Keene et al., 2007). As we have no information about the size of the INPs, except that they are larger than $0.2\,\mu$m, we cannot say what enrichment factor would be

an appropriate assumption in regard to INPs, but the above mentioned literature indicates that processes exist that can produce sufficiently high enrichment factors at least for some substance classes. But it should be also noted, that the laboratory study by (Ickes et al., 2020) did not find a correlation between total organic carbon content of algal culture samples and the freezing of the sample. The same study confirmed that the transfer of ice nucleating material from the seawater to the aerosol phase can indeed happen. Therefore a marine source for the INPs in the Arctic atmosphere cannot be ruled out, but considerable

enrichment of INPs during the transfer from the ocean surface to the atmosphere would have to take place.

## 4 Case Study

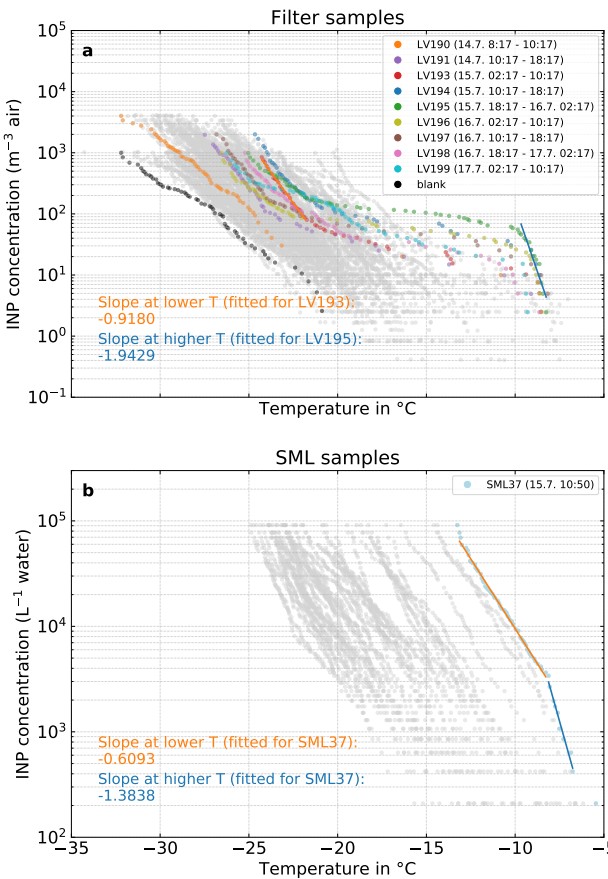

**Figure 11.** a) Filter samples with those the ones corresponding to the case study highlighted. Exemplary fits for the slopes at lower and higher temperatures for case study related samples are shown as orange and blue line respectively. b) SML samples with the case study related sample SML37 shown in light blue. As in a) the fits of the slope are shown as orange and blue line.





In section 3.1, we described that INP concentrations are different in the ice free ocean, within the ice pack, and close to land. In the following we will show, that merely the proximity to land does not make marine INP sources inferior to terrestrial ones. To elucidate this, we consider a time period of several filter sampling intervals which occurred around a time when both, atmospheric and INP concentrations in the SML, were highest. This happened close to Svalbard and in the vicinity of the ice edge, which makes the situation even more interesting.

The overall most ice active SML sample, *SML37*, was taken on July 15th, 10:50, and is highlighted in Fig. 11 (lower panel, light blue symbols). It occurs at the beginning of the sampling period of *LV194*, the second most ice active atmospheric filter sample (Fig. 11, upper panel, blue symbols). A number of atmospheric samples collected before and after sample LV194 are also shown. Most of the $N_{\mathrm{INP}}$(T) spectra from these samples have a very similar overall shape, featuring a fairly steep increase at temperatures above -10 °C, followed by a plateau region between ca. -10°C and -21°C and another, but less steep increase below -21°C. Such a behaviour is indicative for the presence of distinct INP populations and therefore not many mixing events happened during transport (Hartmann et al., 2020; Welti et al., 2018). Additionally, the INPs active at these warmer temperatures are likely biogenic and proteinaceous as indicated by heat tests described in section 3.1.

Temperature range and slope of, e.g., the initial increase in the temperature spectra are somewhat characteristic for the INPs prevailing. In other words, similar slopes in similar temperature ranges observed for atmospheric and SML samples could be indicative for similar INPs being present in both compartments (Knackstedt et al., 2018). As shown in Fig. 11, the slope of the atmospheric samples at T > -10°C (linear fit on logarithmic axis) is -1.94, while the slope of the SML sample at T > -8.2°C is -1.38. The difference in these slopes is too large to unambiguously attribute both samples to the same INP species, and too small to reject the possibility. In other words, the similarities in the spectra (temperature range and slopes) at the high freezing temperatures do not prove but can be taken as a hint at similar INPs being present in both the atmosphere and the SML. It is also apparent that the less steep slope at lower temperatures (T < -8.2°C) of that SML sample has no counterpart in the atmospheric samples. If the atmospheric INPs active above -10°C would originate from the ocean, this suggests that the aerosolization process might be different for different INP species. Furthermore, it is possible that the INP flux is the other way around, i.e. INPs from the atmosphere are deposited into the SML. However this is highly speculative and needs further research.

To further elucidate the possible connection between atmospheric INPs and INPs in the SML, in the following we consider additionally available aerosol related and meteorological information.

The highly ice active sample discussed above, *SML37*, was taken during a period (approx. 14.07. 18:00 to 15.07. 19:30) which, as can be seen in Fig. 12 was characterized by a monomodal particle size distribution and, compared to the periods before and after, increased total particle number ($N_{\mathrm{total}}$,panel b) and CCN ($N_{\mathrm{CCN}}$, panel c) concentrations. In panel d) it can be seen that during this period the wind speed decreased significantly and the wind direction changed slowly from around 240° to 175°. Furthermore, during the collection times of the filter samples *LV194* and *LV195* increased Chlorophyll-a concentrations were measured by the vessel's Ferrybox system. The elevated Chlorophyll-a concentrations may indicate enhanced biological activity like a phytoplankton bloom in the vicinity of the vessel during the collection time of the samples *LV194* and *LV195*. Interestingly the Chlorophyll-a concentration of sample *SML37* is not unusually high (0.24 µg/L Bracher, 2019). This may have two main reasons. Firstly, while Chlorophyll-a is an indicator for biological activity, not all marine microorganisms con-



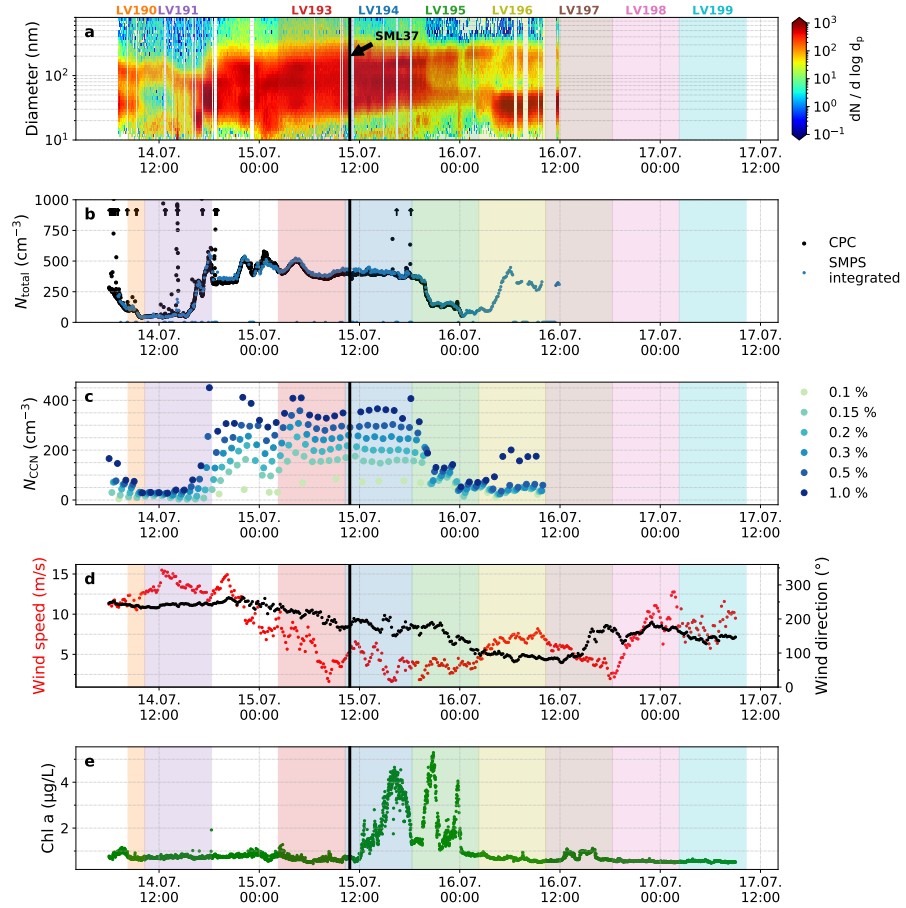

**Figure 12.** Aerosol measurements and other relevant parameters during the period of the case study. Shown are a) the particle number size distribution, b) the total particle concentrations, c) cloud concentration nuclei concentration for 6 different supersaturations, d) 10-minute averages of wind direction and wind speed, and e) Chlorophyll-a concentration measured by the Ferrybox system of the Polarstern. The arrows in panel b) indicate where total particle concentrations are higher than the axis limit. The colored shaded areas mark the periods of where the filter samples were collected. The respective sample ID is shown on top. The black vertical line marks the collection time of the SML sample *SML37*.

tain Chlorophyll-a. Secondly, since Chlorophyll-a itself is not the INP, a correlation is not necessarily to be expected. Also as described in Zeppenfeld et al. (2019) and the references within, the release of ice active algal exudates may be a feature of decaying plankton blooms. Hence the peak in biological activity, indicated by the Chlorophyll-a concentration, may be already

over, when the peak concentration of ice active substances occurs.

To broaden the perspective beyond the aforementioned measurements at the position of the ship itself, HYSPLIT back trajectories were also assessed. In Fig. 13 the hourly 3-day backtrajectories (50 m arrival height) for the entire period depicted in Fig. 12 are shown. The colour code indicates into which collection time, i.e., sample the respective trajectories fall (corre-





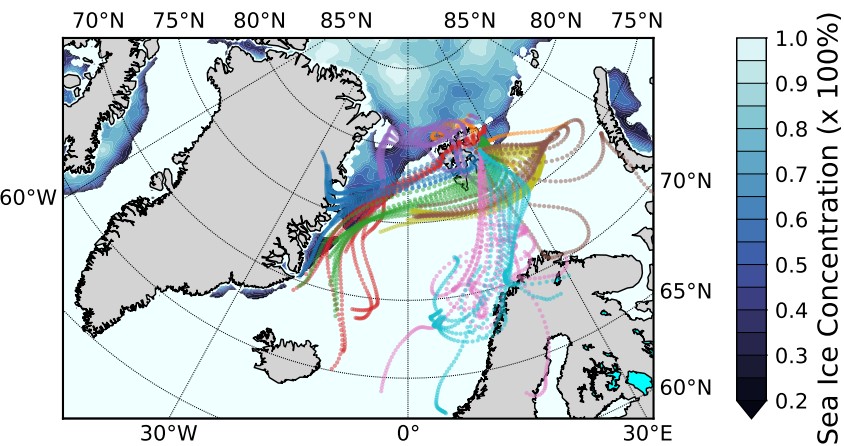

**Figure 13.** Hourly 3-day back trajectories (50 m arrival height) for the collection time of the filter samples *LV190* to *LV199*. The color code shows to which sample the trajectory belongs (consistent with Fig. 11 and 12). The SSMIS sea ice concentration with emphasized ice edge on 15 July 2017 is also shown (Sea ice concentration product of the EUMETSAT Ocean and Sea Ice Satellite Application Facility).

sponding to background colors used in Fig. 12). The trajectories can be categorized into four clusters. The first cluster consists
of the trajectories belonging to the samples LV190 and LV191 (orange and purple). These trajectories travel mostly over the
ice pack north of Svalbard and have no connection to land. The second cluster comprises of the samples LV193, LV194 and
LV195 (red, blue and green), for which the air masses were at the East coast of Greenland before traveling along the ice edge
and South of or over Svalbard, before reaching the ship. While CCN and INP number concentrations were elevated during the
phase indicated by the red and blue trajectories, the phase connected to the green trajectory coincides with a strong lowering of
CCN but still high INP concentrations. The third cluster, which consists of the samples LV197 and LV198 (yellow and brown),
came from the same direction as the second cluster, but made an additional loop towards the East and back, for which it took
about 1 day. For these, $N_{INP}$ were still high, with medium high concentrations of $N_{CCN}$ and $N_{tot}$ during the yellow phase.
Unfortunately, aerosol characterization measurements were no longer continued after that time. The trajectories of the fourth
cluster (LV198 and LV199; pink and cyan) come from the South and had contact with the Norwegian coast 1-2 days prior to
their arrival at the Polarstern. Additionally, Fig. S5 in the SI shows a map of the wider investigation area together with satellite
measurements of Chlorophyll-a concentrations and the backtrajectories, where it can be seen that biological activity can be
found in the region.

While it is a reasonable assumption that the INPs are produced in situ by biological processes, a recent publication by Corn-
well et al. (2020) presents a different pathway. They show in a laboratory study that mineral dust deposited in seawater can





be re-aerosolized and add to the atmospheric INP population. This pathway is especially interesting as a pathway, because in proximity to the coast melt water streams can transport dust into the ocean. Pfirman et al. (1989) also decribe that sea ice often contains dust particles that can originate from atmospheric deposition, or from shelf or shorefast sea ice which may be transported away from the coast. And as Tobo et al. (2019) observed, dust can be also the carrier for highly ice active biogenic material, and therefore these dust related-processes may also explain spatially confined areas of high ice activity without being contradictory to the assumption of biogenic INP.

Summarizing and interpreting our observations it can be stated that:

- The freezing spectra of atmospheric INPs are similar in shape, indicating that during the case study similar atmospheric INP populations were sampled.

- Heat tests indicate that INPs active above -15 °C are biogenic and proteinaceous.

- The freezing spectra of atmospheric INPs and INPs from the SML feature similar slopes at temperatures above -10 °C, suggesting a connection between both compartments, which, however, as discussed above, would need a substantial enrichment of INP during the sea spray production..

- Aerosol particle parameters show that clearly different air masses arrive at Polarstern over the course of the case study.

- Backtrajectories indicate that sampled air masses have different regions of origin and travel over different pathways towards Polarstern.

- Elevated Chlorophyll-a concentrations were observed for a short phase directly at at the position of Polarstern (Ferrybox) and also in the wider geographical region in the week-long satellite composite. This indicates a high biological activity in the investigation region.

We interpret these findings as strong indication for a local marine source being present during our case study. Seemingly this is in contradiction to the results gained from the analysis of fog-water as presented above, unless a significant enrichment of INPs takes place during the aerosolization of seawater and/or SML material. In other words, there is a strong need for gaining knowledge concerning the mechanisms of aerosolization and resulting fluxes of INP and related species at the ocean-atmosphere-interface.

## 5   Summary and Conclusions

We present the results of INP-related investigations carried out during a two month cruise (May - July, 2017) on the RV Polarstern in the Arctic. Four different compartments, i.e., air, fog water, sea surface microlayer and bulk seawater were sampled. Concerning air sampling, throughout the whole cruise, 8 hour filter samples for off-line INP analysis in the TROPOS laboratories were taken and a continuous flow diffusion chamber provided online INP data. Fog samples were collected on an event basis, while samples from the SML and the bulk seawater were taken daily.





The time series of atmospheric $N_{\mathrm{INP}}$ derived from filters show that $N_{\mathrm{INP}}$ was low when the ship was located beyond the ice edge within the ice pack. Higher concentrations were observed outside the ice pack in the MIZ and the ice free-ocean. The highest INP concentrations occurred between both legs of the expedition, when Polarstern was near Longyearbyen harbor and when the vessel cruised along the MIZ at the eastern coast of the Svalbard archipelago at the end of the second leg. For most of the air samples freezing was initiated below -15°C, however some samples featured freezing onsets at warmer temperatures of

up to -7°C. Heat tests suggest the presence of biogenic, proteinaceous INPs at temperatures above -15°C. At -32°C the Arctic $N_{\mathrm{INP}}$ we observed are in the same order of magnitude as in the outflow region for mineral dust from the Saharan desert west of Africa, which indicates that at these low temperatures dust is an important INP even in the Arctic.

SML samples from the biologically active MIZ have a higher fraction of highly ice active samples than the other ocean compartments (ice-free ocean, ice pack, melt pond). Besides that the ice activity of SML and BSW samples is not simply

correlated with the environment the sample was taken from. In general, few highly ice active samples stand out against the other samples. Except for one case, we found the SML to be weakly to significantly enriched in INPs compared to the underlying BSW. The enrichment factors (EF) varied between close to 1 and 94.97 at -15°C. The most enriched samples featured the highest ice activity.

From INP concentration in the fog water and the measured CCN number concentrations we derived potential $N_{\mathrm{INP}}$ in the air,

which we compared to the directly measured $N_{\mathrm{INP}}$, and found good agreement. This indicates that the same, or at least similar, INP populations were present in corresponding fog water and air samples, suggesting that during fog events INPs are activated to droplets and become available as immersion nuclei inside the fog droplets.

Using the ratio of NaCl mass concentration in the air and in the seawater as a scaling factor, we assessed if atmospheric $N_{\mathrm{INP}}$ can be explained simply by aerosolization of SML and BSW material. At any given temperature we found SML- and

BSW-derived $N_{\mathrm{INP}}$ to be 4 to 5 orders of magnitude lower than the $N_{\mathrm{INP}}$ directly measured in air. This clearly shows that aerosolization of SML or BSW material, without significant enrichment of INPs during aerosolization, does not suffice to explain $N_{\mathrm{INP}}$ in air. In other words, a marine source for the INPs in the Arctic atmosphere is possible, but enrichment of INPs by several orders of magnitude during the transfer from the ocean surface to the atmosphere has to take place.

In a case study we looked more deeply into a scenario for which coinciding SML and air samples were highly ice active.

Thereby, we found similarities in the temperature spectra of the highly ice active INPs in the SML and in the air. Air mass changes, indicated by changes in aerosol properties and back trajectories, did not cause changes in the observed INP population. Isolated patches with chlorophyll-a concentrations of about one order of magnitude higher compared to their surroundings underline high biological activity in the investigation region for the time period we investigated in the case study. We consider this as indications for a local biogenic marine source of INPs being present.

Altogether, we found INP concentrations in air, fog water, SML and BSW to be highly variable, with a small number of cases featuring significantly enhanced ice activity. This emphasizes the episodic, highly variable nature of INPs as it was already described decades ago by Bigg (1961). This puts a question mark to the appropriateness of parameterizations based on aerosol particle number in atmospheric models. We found indications for a marine biogenic INP source, however further investigations are needed to gain quantitative knowledge concerning the aerosolization process and the resulting INP fluxes at the interface





between the atmosphere and the ocean surface.

Lastly, to take up the questions from the introduction:

– *What is the abundance of Arctic INPs and in what temperature range can they nucleate ice?*
We found INP active between -7°C and -38°C over a concentration range from $4 * 10^{-1}$ m$^{-3}$ to $1 * 10^{8}$ m$^{-3}$. Most of the time $N_{\text{INP}}$ was at the lower end of $N_{\text{INP}}$ range known from mid-latitudes or even lower. Exceptions were the upper and

lower end of the temperature range: At -10°C $N_{\text{INP}}$ of up to $6 * 10^{1}$ m$^{-3}$ were observed, while at -32°C $N_{\text{INP}}$ was in the same order of magnitude ($10^{5}$ m$^{-3}$) as in the outflow region of the Saharan desert.

– *What is the nature of Arctic INPs (biogenic material vs. mineral dust)?*
We find indications that the warmer temperatures (>-15°C) are dominated by biogenic INP, while at colder temperatures (<-25°C) likely mineral dust dominates.

– *What is the origin of Arctic INPs (local vs. long range transport, marine vs. terrestrial)?*
For the INP at warmer temperatures we find indications that they are marine and locally emitted, which, however, necessitates an enrichment of INP during sea spray aerosol production of several orders of magnitude..

*Data availability.* Freezing spectra are made available at PANGAEA: Link will be added during review





*Competing interests.* The authors declare that they have no conflict of interest.

*Acknowledgements.* We gratefully acknowledge the funding by the Deutsche Forschungsgemeinschaft (DFG, German Research Foundation) – Projektnummer 268020496 – TRR 172, within the Transregional Collaborative Research Center "ArctiC Amplification: Climate Relevant Atmospheric and SurfaCe Processes, and Feedback Mechanisms (AC)³". We thank Susanne Fuchs for the ion chromatographic measurements. We also thank Amelie Assenbaum, Audrey Brown, Mareike Löffler and Jasmin Lubitz for their assistance with the cold stage measurements.



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
