# Peer review of "Terrestrial or marine? – Indications towards the origin of Ice Nucleating Particles during melt season in the European Arctic up to 83.7°N"

_Atmospheric Chemistry and Physics, 2020_

## Referee Comment (RC1) · Anonymous Referee #1 · 4 Jan 2021

Hartmann et al. present findings from shipborne INP measurements during the late spring/early summer of 2017 in the European Arctic. The study is very comprehensive, and I appreciate the targeted scientific questions presented and addressed by the research on aerosol, SML, and BSW INP analyses in tandem with supporting aerosol, meteorological, oceanographic, and air mass modelling data. The authors provide a detailed account of observations and closure resutls from the entire campaign while ultimately focusing on a case study linking SML and airborne INPs. On the flip side, there are some substantial weaknesses that should be remedied. This paper is certainly suitable for publication after the comments below are addressed.

**Major comments:**

I appreciate how the authors introduced scientific questions that cover crucial gaps in Arctic INP research. However, they are indeed quite broad and cannot solely be answered by one study in one region during one time period. They are indeed a fantastic objective, but the authors should be clear that their results provide insight into these questions, i.e., specifically for the European Arctic in the summer. These findings are likely not consistent with other locations and certainly, not other seasons (e.g., the spring Arctic haze season when long-range transport is prominent or winter polar night where open water and biological productivity are at their minima).

The combination of LINA and INDA is very useful, however, it is not clear why: (1) both techniques are not presented per sample even though the methods imply sample prep for both was executed and (2) one technique was presented over the other in the figures instead of both. For example, are Fig 2 and 8 INDA or LINA data? Why were INDA data not shown for Fig 3 and LINA data not shown for Fig 5? Perhaps this issue would be resolved if the authors articulated which offline technique was conducted for which samples and why.

Were the blank spectra subtracted from the filter sample data? The only place I see a blank spectrum is in Fig 11a, but is this one filter or an average of all the blanks collected as indicated in the methods? And is this blank shown from the filter in ultrapure water or ultrapure water alone? Because the background for INPs can widely vary for the type of ultrapure water used, a bit more detail on the source of the ultrapure water used should be provided (e.g., Milli-Q, DI, etc.). Blanks for both plain ultrapure water and the blank filters in ultrapure water should be shown to demonstrate the reliability of the ultrapure water and filters used. Additionally, DI blanks can be all over the board depending on the container used for sampling things like seawater. Was a blank conducted for putting ultrapure water in similar sampling containers as the SML, BSW, and fog water were collected in? If so, these blanks should probably be shown in the SML, BSW, and fog water spectra as well (at least, the first occurrence of each).

Frankly, it is a bit difficult to discern the generalized difference in the offline INP spectra from ice-free ocean, ice pack, and the MIZ for the aerosol (e.g., in Fig 3, perhaps this is somewhat clear > -15C, but not the bulk of the spectra) and those + melt ponds for the SML and BSW (e.g., Fig 5 has highest spectra from the ice pack and MIZ for SML; and ice-free ocean, melt pond, and ice pack for BSW, depending on the temperature). To support the authors' claims regarding which regions have the highest INPs, perhaps a figure summarizing the data would be useful. For example, a figure showing: (1) aerosol, (2) SML, and (3) BSW INP concentrations at select temperatures (e.g., =10, - 15, -20, etc.). Then, this would clearly demonstrate which indeed had higher concentrations and at which freezing temperatures.

Where were the SML and BSW samples collected from in the ice pack, if not from melt ponds? Were these samples collected in leads I assume?

The authors state several times that the heat tests suggest the presence of biogenic INPs, but this is generalized without any discussion on how this statement came to be. In looking at the SI figure, indeed some spectra show a decrease in INPs after treatment, but it is difficult to tell if this is the case for all samples at all temperatures. In the results and discussion when first mentioned, the authors should elaborate briefly, meaning providing a more quantitative assessment of the decrease (how much of a decrease exactly?), if this was consistent for all samples and temperatures (was it?), and why or why not.

I assume this is because it would overcrowd the figures, but why are uncertainty bars not shown on any of the spectra, aside from the fog water derived INPs? Is there a way to possibly show uncertainty to corroborate the statements regarding differences between the different sample types?

The case study is indeed interesting but could be described in more detail beyond the one SML37 sample. For instance, what did the BSW spectra look like during this time period? If there was some level of blooming happening, that should be evident in the bulk seawater more so than the SML since phytoplankton reside in the upper trophic levels of the ocean. From the limited body of work on blooms and linkages to INPs, we would not expect chl-a to correlate directly with INPs (e.g., work by Creamean et al., Irish et al., McCluskey et al., and Zeppenfeld). It is more likely that INPs increase following a bloom due to enhancement in biogenic byproducts and bacterial growth following the peak of the bloom, versus INPs originating from the phytoplankton, especially considering phytoplankton have been shown to serve as only moderately effective INPs at colder temperatures. Looking at Fig 12, particularly at temps > -15C, it is interesting how LV195 had the highest warm-temperature "bump" of INPs followed by LV196 and LV197 towards the end of and following the spike in chl-a, while LV194 was also relatively high into the first half of the spike. It certainly would be interesting to highlight and discuss any SML samples following this period in addition to BSW samples before, during, and after, for comparison. Also, it looks like some of the relatively warmer temperature SPIN data were high at the beginning of this event (-26 to -32C), but it is difficult to tell from Fig 2. To provide a nice, holistic wrap up by presenting all the data during the case study, showing and discussing the SPIN data would be useful as well.

Along these lines, the comparison of the slopes is a bit tenuous, given the slope changes could be cause by numerous factors including but not limited to sample-to-sample variability, influences from other airborne sources as indicated, etc. etc. In my opinion, the slope analysis does not add value to the results and are not as convincing as, say, the relationship between bloom time and INPs described in my previous comment. I suggest eliminating the slope discussion and focusing more on comparing the increases and decreases in the INP concentrations between the water and air in the context of the supporting measurements (chl-a, NC, NCCN, etc.). These are what provide empirical evidence of INP sources instead of arbitrarily defining slopes for only select spectra.

The authors might consider reordering the sections in the results and discussion. As it stands, this section does not flow as well as it could, and it is not clear why the case study is section 4 when it is still a part of the results and discussion (section 3). This inherently causes some of the more important findings are buried. It makes more sense to order in the following as a combined section 3 of results and discussion (note that the titles do not need to be exactly these, but something along the lines). First, describe the sources: 3.1 Atmospheric INP sources over the entire campaign (this should

include low temperature as a paragraph and not a subsection) and 3.2 INPs in the SML and BSW over the entire campaign. Next, describe linkages between the sources and airborne INPs: 3.3. connecting to sea spray. Then, possible effects on fog, which includes measured and derived: 3.4 Measured and derived fog INPs. Last, focusing on a unique case: 3.5 INPs from a phytoplankton bloom.

The conclusion that dust represents the cold temperature INPs solely based on the transect in Fig 4 is very tenuous. Dust is also present at temperatures warmer than -32C, so why is this generalization based on only one temperature? How can the authors be sure what they measured were not algal or diatom material INPs, which do glaciate at the same range as dust? What kind of drop in concentrations occurred in the heat treatments at these low temperatures? Can the authors link the air mass transport pathways to any major dust source like the Sahara? It does not look like it in Fig 13, especially considering how much time the air masses spent over water as compared to land. Certainly, aerosolized redistributed dust from the ocean surface could have been a possible source, but there is no evidence of this specific to this study region. From the evidence provided, the authors should not rule out other possible INP sources at these low temperatures, especially without proof of air masses interacting with dust sources along their transect, which would entail a more detailed analysis that might include remote sensing data.

**Minor comments:**

Lines 21-22: The statement on heat is redundant from earlier in the abstract; should remove.

Lines 111-112: How long were samples stored until analysis?

Lines 229-234: Since the online measurements are a primary focus of the manuscript (i.e., not supporting), they should maintain their own section in the methods, perhaps before or after the offline INP measurement section.

Fig 2: Why are SPIN data missing from the very beginning of the campaign? If these data were not obtained or missing, that should be noted in the caption or figure itself.

Lines 277-278: Which "others"? This statement is a bit vague.

Lines 281-283: This is not clear given the 8-h and 2-h samples are not visually distinguished in Fig 3. Perhaps use of different markers for each would help?

Fig 6: What does it mean by "stands out in terms of ice activity"? Was this qualitatively or quantitatively determined? Either way, there should be some description of how this was defined.

Fig 11: Clarify in the caption that the grey spectra are those from the entire study.

---

## Referee Comment (RC2) · Anonymous Referee #2 · 5 Jan 2021

In the paper "Terrestrial or marine? - Indications towards the origin of Ice Nucleating Particles during melt season in the European Arctic up to 83.7°N" by Hartmann et al. 2020 the authors present field measurements of INP in the European Arctic across different temperature ranges and try to investigate the sources of these measured INP. To do so additional measurements of INP in bulk sea water, the sea surface microlayer and fog water are done, (satellite) measurements of Chlorophyll-a and air mass history (trajectory model) are analysed. The measurements and analysis are detailed and thoroughly done. Therefore the study is suitable for ACP after some minor revisions.

General comments:

- The paper does and can only answer the INP source question in a very limited way. Even though a lot of different measurements/analysis is used, the conclusions are a bit vague, which is probably due to the complex nature of the research question. They find indications that INP above -15°C are dominantly biogenic, while at lower temperatures dominantly dust (this is not so surprising looking at previous field studies, even though not so many specific studies for the Arctic exist). They state that the sources could be marine, locally emitted for the INP above -15°C, but only if the INP are enriched during aerosolisation by several orders of magnitude. It would be helpful to discuss more critically the limitation of the study, what can be taken from it for which conditions and what needs further investigation. The authors might need to break down the complex research question into subquestions etc..

- Why are two different filter techniques (LINA and INDA) used and what are the advantages/disadvantages of both compared to each other? When is which technique used (figure captions are not always clear on that?)?

- It is not clear where the conclusions come from that the lower temperature INP are dominated by dust (no clear indication from the trajectories?). Is that derived from the slope of the measurement and comparison to other field measurements?

- Section 3.3.1: How large is the variation due to the assumptions made (SS etc.)? Could you add a section on commenting this and or add error ranges depending resulting from the assumptions made? Same for the variable N_CCN which might affect the result (and comes itself already with an uncertainty).

- How large are the uncertainties in the calculated trajectories? Please comment on limitations here as well.

Specific comments:

- Page 2, line 21 and 22: The sentence on the heat induced reduction... can be removed, this was stated already on page 1, line 9 ff.

- Page 2, line 35-46: It would make sense to switch paragraph line 35-39 with the paragraph line 40-46.

- Page 5, line 127: "within the ice pack" = open leads?

- Page 9, line 246: What is a "vessel's Ferrybox system"? How does it work?

- Page 12, line 290-291: Was the sampling time consistent for all environmental settings or how can that influence the statement?

- Fig. 4: Is that figure needed in this paper?

- Fig. 7: The labels are a bit small here.

- Page 15, line 340: Add: "...can be seen" due to filtration.

- Page 16, line 354-355: Is it legitimate to compare sea water and fog water directly? Is there not a bias due to dillution effects?

- Fig. 11 a: The fit/slope at lower T is very difficult to spot in the graph.

- Fig. 12: Write variables in the caption.

- Fig. 12: The arrows are very small to spot.

- Page 24, line 503: Be more specific what "similar in shape" means.

- Page 25, line 537-538: The most enriched samples featured the highest ice activity. -> that refers to the SML samples?

Technical corrections:

- Page 11, line 270: Remove brackets around the temperatures.

- Page 14, Caption Fig. 5: ...the samples were taken from.

- Page 14, Caption Fig. 5: Add a . at the end of the caption.

- Page 20, line 432: Remove the brackets around the citation.

- Page 21, line 465: Add a space between N_total, and panel b.

- Page 21, line 470: Add a ; between 0.24... and the reference Bracher, 2019.

- Page 24, line 511: Remove one at.

- Page 26, line 572: Remove one . at the end of the line.

---

## Author Comment (AC1) · 5 Apr 2021

We thank referee 1 for reviewing our manuscript and for giving comments and suggestions. Our answers are given in blue, below, while the original text of the review was kept in black.

Referee #1 Evaluations:
**Scientific significance**: Good
**Scientific quality**: Excellent
**Presentation quality**: Good

Major comments:

I appreciate how the authors introduced scientific questions that cover crucial gaps in Arctic INP research. However, they are indeed quite broad and cannot solely be answered by one study in one region during one time period. They are indeed a fantastic objective, but the authors should be clear that their results provide insight into these questions, i.e., specifically for the European Arctic in the summer. These findings are likely not consistent with other locations and certainly, not other seasons (e.g., the spring Arctic haze season when long-range transport is prominent or winter polar night where open water and biological productivity are at their minima).

We agree that a single, two months field campaign cannot answer all open questions regarding Arctic INP. We think that already the title puts into perspective that we show results for a specific season and region. In the revised version of the manuscript we added an additional sentence to highlight that our findings are constrained by season and region: "[…] Our findings are constrained to the season and region in in which measurements took place. They will nevertheless contribute to a better understanding concerning Arctic INPs and their potential effects on Arctic clouds.[…]" (L81 – L82)

The combination of LINA and INDA is very useful, however, it is not clear why: (1) both techniques are not presented per sample even though the methods imply sample prep for both was executed and (2) one technique was presented over the other in the figures instead of both. For example, are Fig 2 and 8 INDA or LINA data? Why were INDA data not shown for Fig 3 and LINA data not shown for Fig 5? Perhaps this issue would be resolved if the authors articulated which offline technique was conducted for which samples and why.

Which instrument was used was constrained by instrument availability. While the filter samples were scheduled to be analyzed, INDA was not continuously available, hence a full dataset for these exists only from LINA. For the sea water samples LINA and INDA measurements are available, and are now also shown in Fig, 5. In general, INDA measurements are more sensitive to low INP concentrations at warm temperatures that are relevant for the biological INP and also provide better signal to background ratio (sample in comparison to pure MilliQ water). Hence, data measured with INDA was preferred over LINA were possible.

In the revised version of the manuscript, we now indicate which instrument was used in the figures showing freezing spectra (Fig. 2, 3, 5, 6, 7, 8, 10 and 11).

Were the blank spectra subtracted from the filter sample data? The only place I see a blank spectrum is in Fig 11a, but is this one filter or an average of all the blanks collected as indicated in the methods? And is this blank shown from the filter in ultrapure water or ultrapure water alone? Because the background for INPs can widely vary for the type of ultrapure water used, a bit more detail on the source of the ultrapure water used should be provided (e.g., Milli-Q, DI, etc.). Blanks for both plain ultrapure water and the blank filters in ultrapure water should be shown to demonstrate the reliability of the ultrapure water and filters used. Additionally, DI blanks can be all over the board depending on the container used for sampling things like seawater. Was a blank conducted for putting ultrapure water in similar sampling containers as the SML, BSW, and fog water were collected

in? If so, these blanks should probably be shown in the SML, BSW, and fog water spectra as well (at least, the first occurrence of each).

We did not subtract blanks from real samples. As ultrapure water we used Milli-Q. This information is added in the revised manuscript ("[…] ultrapure water (Type 1; Direct-Q3 Water Purification System, Merck Millipore, Darmstadt, Germany) [...]"; L154-L155). The blank in figure 11 is the average of the field blank filters taken during the campaign. That information was added to the capture of Fig 11: "The black dots depict the mean freezing spectrum of the field blanks scaled to atmospheric concentrations with the mean sampled air volume of the 8h filter samples."

Prior to each measurement day, we typically perform a test of the Milli-Q water that is used on that day, in order to notice possible problems. For the data shown it can be excluded that contamination of the Milli-Q water (above the "usual level") is an issue. Also contamination of the containers that were used can be excluded: In the field, SML, BSW and fog water samples were collected into Nalgene bottles pre-cleaned with HCl solution (10% v/v) and then divided into two aliquots for the saccharide analysis of Zeppenfeld et al. (2019) and the INP analysis of this manuscript. The aliquot for the INP analysis was stored in a sterile centrifuge tube (50 mL, Cellstar, Greiner Bio-One, Kremsmünster, Austria), which is the same kind of container in which the washing of the filters was performed. These centrifuge tubes are used for all INP measurements of filter samples in our laboratory  and so far contamination by the container was not observed. Therefore, we are confident that our typical background always is valid. Where applicable freezing spectra of pure MilliQ water and field blanks are added to figures (Fig. 3, 5, 8).

Frankly, it is a bit difficult to discern the generalized difference in the offline INP spectra from ice-free ocean, ice pack, and the MIZ for the aerosol (e.g., in Fig 3, perhaps this is somewhat clear > -15C, but not the bulk of the spectra) and those + melt ponds for the SML and BSW (e.g., Fig 5 has highest spectra from the ice pack and MIZ for SML; and ice-free ocean, melt pond, and ice pack for BSW, depending on the temperature). To support the authors' claims regarding which regions have the highest INPs, perhaps a figure summarizing the data would be useful. For example, a figure showing: (1) aerosol, (2) SML, and (3) BSW INP concentrations at select temperatures (e.g., =10, - 15, -20, etc.). Then, this would clearly demonstrate which indeed had higher concentrations and at which freezing temperatures.

We are not sure if we understand the referee's suggestion for a figure correctly, but in order to improve the discernibility we changed/added two things in the revised manuscript:

1. we adjusted the color-coded background in Fig. 2, the time series of $N_{INP}$, to match the categorization of the samples.

2. we created a box plot of the data shown in Fig. 3 for selected temperatures (-10°C, -15°C, -20°C and -25°C). The box plot is shown below.

[Figure]

Regarding the boxplot, the way we treated values outside the detectable range has to be kept in mind: In offline INP analysis, $N_{INP}$ cannot be derived for $f_{ice} = 0$ and $f_{ice} = 1$, i.e., when no or all droplets of an experiment are frozen respectively. Therfore, the lower and upper limit of detection (LOD) is given by $N_{INP}(f_{ice} = 1/n_{total})$ and $N_{INP}(f_{ice} = (n_{total} - 1)/n_{total})$ respectively (with $n_{total}$ = the number of droplets used in the experiment). How the LOD is treated, when calculating statistics such as mean, standard deviation, etc. directly influences if the used statistic is over- or underestimated. For example: If the lower LOD is (i) treated as the absolute value, the mean would be overestimated; (ii) treated as zero, the mean would be underestimated; (iii) excluded from the data set, the mean would be overestimated; (iv) substituting the LOD with a value between zero and the LOD, may lead to over- or underestimation of the mean. To our knowledge, for offline INP measurements no established way for the treatment of the LOD in such cases exist. Hence, we adapt the way of treatment which is widely used in environmental chemistry (U.S. EPA, 2020) and substitute the lower LOD with LOD/2 and the upper LOD with LOD*2 in order to calculate the statistics which are also shown in the box plot. In the same manner box plots for the SML and BSW data were created:

[Figure]

We also want to emphasize the point that we did not claim in the manuscript, that one of the environmental settings has generally higher INP concentrations, but rather that the most ice active samples (at warmer temperatures) are more often associated with the MIZ than the icepack (for filter and SML samples). Hence, in terms of the boxplot, not only the box itself, which represents the quartiles, but also the fliers have to be considered.

The box plots shown above were added to the SI with the following description: "Fig. S10 shows a box plot of the filter derived $N_{INP}$ shown in Fig. 3 of the main manuscript. The horizontal line represents the median and the green triangle the mean. The whiskers have the length of 1.5 * IQR and data points outside the range of the whiskers is shown with diamond markers. It should be noted that for samples whose value was outside the detectable range at the selected temperature, the value was substituted in order to minimize the over- and underestimation of the summary statistics needed to create the box plot. If at the selected temperature a sample had an $N_{INP}$ value below the detectable range, the value was substituted as LLOD/2 (LLOD = lower limit of detection). Analogous values above the detectable range were substituted with ULOD*2 (ULOD = upper limit of detection) Box plots for the SML and BSW samples were created in the same manner as described before (Fig. S11 and S12; same data as in Fig. 5 in the main manuscript)." (L41-48)
We refer to these figures in the main manuscript: "Additionally, Fig. S10 in the SI shows a box plot of the very same filter samples, in order to emphasize the general differences between the environments." (L284-285) and "Additionally, Fig. S11 and S12 in the SI show box plots of the very same SML and BSW samples in order to emphasize the general differences between the environments." (L332-333)

Where were the SML and BSW samples collected from in the ice pack, if not from melt ponds? Were these samples collected in leads I assume?
Yes, in the open leads within the ice pack. This is now specified in the manuscript: "[…] Seawater samples were taken from different environments: ice-free ocean, marginal ice zone (MIZ), open leads within the ice pack or from meltponds.[…]" (L126-L127)

The authors state several times that the heat tests suggest the presence of biogenic INPs, but this is generalized without any discussion on how this statement came to be.

The heat test is an established method to test for biogenic, proteinacaeous INP and used widely in the literature: Conen et al., 2011, 2012, 2017; Conen and Yakutin, 2018; Felgitsch et al., 2018; Hara et al., 2016a; Joly et al., 2014; Moffett et al., 2018; Hill et al., 2016; Huang et al., 2021; McCluskey et al., 2018; Kunert et al., 2019; Pouleur et al., 1992. The test is based on the heat sensitivity of biogenic, proteinacaeous INP which are denatured at 95 °C, while most other INP are heat resistant.

In the revised version the paragraph now reads as follows: "The test for heat-labile INPs (Fig. S6 and S7 in the SI) demonstrates that ice activity of the samples is reduced when heated for1 h at 95°C. Especially INPs that nucleated ice at temperatures above ca. -16°C are gone after the heating. This is widely seen as an indicator for the presence of biogenic, proteinaceous INPs as those become denatured during the heating, which reduces their ice activity (Conen et al., 2011, 2012, 2017; Conen and Yakutin, 2018; Felgitsch et al., 2018; Hara et al., 2016; Joly et al., 2014; Moffett et al., 2018; Hill et al., 2016;Huang et al., 2021; McCluskey et al., 2018; Kunert et al., 2019; Pouleur et al., 1992)." (L305-310)

In looking at the SI figure, indeed some spectra show a decrease in INPs after treatment, but it is difficult to tell if this is the case for all samples at all temperatures.

We agree. In L306 of the manuscript and L22-23 of the SI, we state that this is not the case for all temperatures. We observe the main decrease at temperatures above -16°C (see Fig. S6 in the SI, showing that no heated sample nucleated ice above -16°C). We argue that the main change is to be expected above -16°C, as biological INP are typically ice active at these higher temperatures. At lower temperatures only a minor decrease occurs. Also in response to the next referee comment below, we created a figure, that shows better how much $N_{INP}$ was decreased after heating at different temperatures (see our response to the next comment below for details).

In the results and discussion when first mentioned, the authors should elaborate briefly, meaning providing a more quantitative assessment of the decrease (how much of a decrease exactly?), if this was consistent for all samples and temperatures (was it?), and why or why not.

For the figure below, we quantified the effect of the of the heat treatment as a decrease in $N_{INP}$ in percent: If $N_{INP}$ of the unheated sample at the selected temperature would be 100 $L^{-1}$ and if after the heat treatment $N_{INP}$ would have been 20 $L^{-1}$, this would be counted as an 80% decrease. Therefore, samples that showed some freezing at the selected temperature before the heat treatment, but no freezing after the heat treatment, are counted as a 100% decrease in $N_{INP}$. This decrease is shown as a boxplot for all samples at selected temperatures. It can be seen that the decrease is most pronounced for the warmer temperatures, but is present throughout the whole temperature range. For temperatures of -16°C and above, the decrease almost always 100%, which indicates that all samples contained heat-labile INP at these temperatures. At temperatures of -18°C and below, the variability in the decrease becomes higher, which indicates that at the lower temperatures some samples still contain mostly heat-labile INP, while in other samples also more heat-stable INP are present.

The figure of the decrease in INP was added to the SI, along with the description: " Fig. S7 shows a box plot of the decrease in $N_{INP}$ after the heat treatment at selected temperatures of all samples that were heated. Boxes represent the 25% and 75% quartile. The horizontal line represents the median and the green triangle the mean. The whiskers have the length of 1.5 * IQR (interquartile range) and data points outside the range of the whiskers is shown with diamond markers. It can be seen that the decrease is most pronounced for the warmer temperatures, but is present throughout the whole temperature range. For temperatures of -16°C and above, the decrease almost always 100%, which indicates that all samples contained heat-labile INP at these temperatures. At temperatures of -18°C and below, the variability in the decrease becomes higher, which indicates that at the lower temperatures

some samples still contain mostly heat-labile INP, while in other samples also more heat-stable INP are present." (L24-31)

[Figure]

I assume this is because it would overcrowd the figures, but why are uncertainty bars not shown on any of the spectra, aside from the fog water derived INPs? Is there a way to possibly show uncertainty to corroborate the statements regarding differences between the different sample types?
In the revised manuscript we added exemplary errorbars to data points in Fig. 3 and 5 to show the range of uncertainty without overcrowding the figures. We also added a paragraph on how the errorbars were derived: "The uncertainty in $N_{INP}$ was calculated with a formula by Agresti & Coull (1998).  Agresti & Coull (1998) published an approximation for binomial sampling intervals, which was applied to $N_{INP}$ measurements by e.g., Gong et al., (2020), McCluskey et al, (2018) and Hill et al., (2016). Following their approach the confidence intervals for $f_{ice}$ are calculated by:

$$\left(f_{ice} + \frac{z_{a/2}^2}{2n} \pm z_{a/2}\sqrt{[f_{ice}(1-f_{ice}) + z_{a/2}^2/(4n)]/n}\right)/(1 + z_{a/2}^2/n),$$

where $n$ is the droplet number, and $z_{a/2}$ is the standard score at a confidence level $a/2$, which for a 95% confidence interval is 1.96." (L208-213)

The case study is indeed interesting but could be described in more detail beyond the one SML37 sample. For instance, what did the BSW spectra look like during this time period? If there was some level of blooming happening, that should be evident in the bulk seawater more so than the SML since phytoplankton reside in the upper trophic levels of the ocean. From the limited body of work on blooms and linkages to INPs, we would not expect chl-a to correlate directly with INPs (e.g., work by Creamean et al., Irish et al., McCluskey et al., and Zeppenfeld). It is more likely that INPs increase following a bloom due to enhancement in biogenic byproducts and bacterial growth following the

peak of the bloom, versus INPs originating from the phytoplankton, especially considering phytoplankton have been shown to serve as only moderately effective INPs at colder temperatures. We mentioned that chl-a does not necessarily have to be related to INP in the manuscript L507-511 (before revision L470-475), citing the work by Zeppenfeld et al. (2019). Typically, we observed either no enrichment or an enrichment from BSW towards SML. Therefore, we refrain from discussing the BSW samples in this context. The spectra of the BSW sample corresponding to SML37 can be seen in the figure associated with our next answer below (lower panel, orange dots). It can be seen, that the high ice activity during the case study period is constrained to the SML. The BSW sample is like the majority of the BSW samples rather unremarkable.

Looking at Fig 12, particularly at temps > -15C, it is interesting how LV195 had the highest warm-temperature "bump" of INPs followed by LV196 and LV197 towards the end of and following the spike in chl-a, while LV194 was also relatively high into the first half of the spike. It certainly would be interesting to highlight and discuss any SML samples following this period in addition to BSW samples before, during, and after, for comparison.
We agree that it would be interesting to investigate SML samples for the period following the case study, but unfortunately July 15 was the last day when SML samples could be collected and no following samples exist. However, for the sake of completeness, please find below a figure where we

[Figure]

highlighted the SML/BSW samples from the case study (orange), and the SML/BSW samples that were taken before the cases study (blue; collected ca. 24 hours prior to SM37/BSW37)

Also, it looks like some of the relatively warmer temperature SPIN data were high at the beginning of this event (-26 to -32C), but it is difficult to tell from Fig 2. To provide a nice, holistic wrap up by presenting all the data during the case study, showing and discussing the SPIN data would be useful as well.

SPIN measurements have been added to Fig.12. For convenience we converted the SPIN data to cm$^{-3}$, which allows easy comparison with $N_{CCN}$ and $N_{total}$ which are also shown in Fig. 12. The SPIN data shows only little variability for a certain temperature and also no trend or correlation with one of the other parameters. The following was added to the manuscript: "Lastly, panel f) in Fig. 12 shows $N_{INP}$ measured with SPIN. Similar to the filter $N_{INP}$, also the INP measurements with SPIN remain fairly constant during the period of the case study and no correlation with the other parameters shown in Fig. 12 can be seen." (L511-513)

Along these lines, the comparison of the slopes is a bit tenuous, given the slope changes could be cause by numerous factors including but not limited to sample-to-sample variability, influences from other airborne sources as indicated, etc. etc. In my opinion, the slope analysis does not add value to the results and are not as convincing as, say, the relationship between bloom time and INPs described in my previous comment. I suggest eliminating the slope discussion and focusing more on comparing the increases and decreases in the INP concentrations between the water and air in the context of the supporting measurements (chl-a, NC, NCCN, etc.). These are what provide empirical evidence of INP sources instead of arbitrarily defining slopes for only select spectra.

We agree that the slope is not an unambiguous indicator. However, from L490 to L496, we discuss the slope comparison thoroughly and point out the speculative nature of that comparison. The central findings regarding the supporting measurements (chl-a, NC, NCCN, etc.) and the INP concentrations is that the supporting measurements change, while the INP remain the same. In that sense we do not see the benefit of discussing these parameters in more detail.

The authors might consider reordering the sections in the results and discussion. As it stands, this section does not flow as well as it could, and it is not clear why the case study is section 4 when it is still a part of the results and discussion (section 3). This inherently causes some of the more important findings are buried. It makes more sense to order in the following as a combined section 3 of results and discussion (note that the titles do not need to be exactly these, but something along the lines). First, describe the sources: 3.1 Atmospheric INP sources over the entire campaign (this should include low temperature as a paragraph and not a subsection) and 3.2 INPs in the SML and BSW over the entire campaign. Next, describe linkages between the sources and airborne INPs: 3.3. connecting to sea spray. Then, possible effects on fog, which includes measured and derived: 3.4 Measured and derived fog INPs. Last, focusing on a unique case: 3.5 INPs from a phytoplankton bloom.

Following the reviewer's suggestion, we restructured some parts of the manuscript.

The conclusion that dust represents the cold temperature INPs solely based on the transect in Fig 4 is very tenuous. Dust is also present at temperatures warmer than -32C, so why is this generalization based on only one temperature? How can the authors be sure what they measured were not algal or diatom material INPs, which do glaciate at the same range as dust?

The assumption that mineral dust dominates cold temperature INP was derived from its abundance and temperature range of activity, based on laboratory experiments. To our knowledge, no study has yet found biogenic INP to be present in such high concentrations as shown in our Fig. 4., the only INP known to occur in such high atmospheric concentrations that it could explain the observed INP concentrations, is mineral dust. Additional confirmation comes from the study by Welti et al. (2020) reporting that they only observed similarly high INP concentrations at -32°C, when they were in the outflow region of the Sahara, where the aerosol is known to be dominated by mineral dust.

However, the referee is of course correct with the comment that also silica-algae like diatoms might contribute to the INP population at the lower temperatures. Therefore we rephrased the paragraph on the cold temperature INP: We still present the argumentation for mineral dust, but also mention diatoms as a source for biogenic, mineral INP. We also now conclude the section with the more general term "mineral INP" rather mineral dust.

The whole paragraph now reads as follows: "In the previous section we described that at warmer temperatures for example at -10°C, samples with high INP concentrations are found more often in the MIZ and less frequently within the ice pack. In comparison, at the lower temperatures measured with SPIN (cross markers in Fig. 4) no correlation with the environmental setting is found. However, in global context the level of $N_{INP}$ at these low temperatures is remarkable by itself as shown in Fig. 4. That figure shows $N_{INP}$ in the Arctic at -32°C measured with SPIN during PS106, but also SPIN data by Welti et al. (2020) of a transect from Bremerhaven (Germany) to Cape Town (South Africa) along the western coast of Africa. It is striking that at these low temperatures $N_{INP}$ in the Arctic are in the same order of magnitude as in the outflow region of mineral dust from the Saharan desert. While we have no means of proofing the presence of mineral dust at these colder temperatures during PASCAL, to our knowledge there are also no other known sources of INP that can produce such high concentrations throughout whole time period of the campaign. Also, it was recently shown by Sanchez-Marroquin et al. (2020) that Iceland can be a strong Arctic dust source. Also Irish et al. (2019a) suggested that observed INP were mineral dust particles originating in the Arctic (Hudson Bay, eastern Greenland, northwest continental Canada), rather than particles originating from sea spray. And global model transport simulations done by Groot Zwaaftink et al. (2016) show that mineral dust is not only transported into the Arctic from remote regions but also, possibly increasingly, generated in the region itself. However, it is also possible also other sources of mineral INP contribute to the INP population at these temperatures. E.g., diatoms represent a biogenic, but mineral source of INP, as they have a cell wall made of silica (Xi et al., 2021) Therefore, it is likely that mineral INP, possibly mineral dust, contribute to $N_{INP}$ at low temperatures during the campaign." (L311-327)

What kind of drop in concentrations occurred in the heat treatments at these low temperatures?
The LINA measurements do not extend to these low temperatures, therefore no additional conclusions can be drawn in this regard.

Can the authors link the air mass transport pathways to any major dust source like the Sahara? It does not look like it in Fig 13, especially considering how much time the air masses spent over water as compared to land.
Extending the back trajectories much further back in time is not meaningful in our view. Kahn (1993) showed that 5-day back trajectories have uncertainties of ca. 1000km. We would also expect dust sources in high latitudes to be more significant than the Saharan desert. It was recently shown by Sanchez-Marroquin et al. (2020) that Iceland can be a strong Arctic dust source. In addition, Irish et al. (2019) suggested that observed INP were mineral dust particles originating in the Arctic (Hudson Bay, eastern Greenland, northwest continental Canada), rather than particles originating from sea spray. Global transport model simulations done by Groot Zwaaftink et al. (2016) show that mineral dust is not only transported into the Arctic from distant regions but also, possibly increasingly, generated in the region itself.

The "dust extinction" (aerosol optical thickness at 550nm) provided by NASA's GEOS-5, shows little variation throughout the whole campaign and no transport from mid- and low latitude dust sources can be seen. For the period of the case study, only on the 14.7.2017 some dust appears to be transported from Novaya Zemlya to the general region north-east of Svalbard. Nevertheless, the dust extinction near the location of the vessel does not change significantly. In general, this is in alignment with our

observations shown in Fig. 4, where $N_{INP}$ is relatively high, but does not vary extensively throughout the campaign.

Certainly, aerosolized redistributed dust from the ocean surface could have been a possible source, but there is no evidence of this specific to this study region. From the evidence provided, the authors should not rule out other possible INP sources at these low temperatures, especially without proof of air masses interacting with dust sources along their transect, which would entail a more detailed analysis that might include remote sensing data.

We agree that we do not have observational proof that mineral dust is certainly the INP at the low temperatures, but to our knowledge no other source could produce high enough concentrations of INP at these temperatures. It also must be kept in mind, that during most of the campaign, the ship was closer than 200km to Svalbard. It seems reasonable to assume that in such close proximity, Svalbard provides an ubiquitous background concentration of mineral dust. This aligns with what we described in the previous answer, where we mentioned that not much variation in the "dust extinction" can be found throughout the campaign.

Minor comments:

The manuscript presents interesting new data on INP concentrations and collocated observations during late winter in the High Arctic. Material and methods were appropriate to produce robust results. Results are clearly presented and discussed. I enjoyed reading the paper.Lines 21-22: The statement on heat is redundant from earlier in the abstract; should remove.

Statement was removed.

Lines 111-112: How long were samples stored until analysis?

The samples arrived in October 2017 at Tropos and where then analyzed over the course of ca ¾ of a year.

Lines 229-234: Since the online measurements are a primary focus of the manuscript (i.e., not supporting), they should maintain their own section in the methods, perhaps before or after the offline INP measurement section.

The paragraph describing SPIN was extended. In the revised manuscript the paragraph now reads as :" In addition to the off-line INPs analysis of the filter samples, also the SPectrometer for Ice Nuclei (SPIN; Droplet Mea-surements Techniques, Boulder, CO, USA) was deployed to measure $N_{INP}$ in immersion mode on-line. SPIN is a continuous flow diffusion chamber (CFDC) with a parallel plate geometry and the measurement principle of SPIN in immersion mode can be briefly described as follows: aerosol particles are activated to cloud droplets and then exposed to conditions where ice can form. The number of formed ice crystals is then optically detected. SPIN is described in detail in Garimella et al. (2016). SPIN was placed within a measurement container. Together with the other aerosol instrumentation was the aerosol was fed to SPIN through one main inlet, but with additional subsequent drying of the aerosol. SPIN sampled in half-hourly intervals of constant temperature and relative humidity and each sampling condition was repeated three times within 24 h. The SPIN dataset is also part of the overview of global ship-borne INP measurements by Welti et al. (2020)" (L236-244)

Fig 2: Why are SPIN data missing from the very beginning of the campaign? If these data were not obtained or missing, that should be noted in the caption or figure itself.

SPIN was not operational in the beginning of the campaign. We added to the caption of Fig. 2: "[…] Note that SPIN measurements were only obtained beginning with May 31."

Lines 277-278: Which "others"? This statement is a bit vague.
It is specified that with "others" we mean the spectra that go up to 1e3 m$^{-3}$.

Lines 281-283: This is not clear given the 8-h and 2-h samples are not visually distinguished in Fig 3. Perhaps use of different markers for each would help?
Due to the large number of samples it is difficult to distinguish the spectra by different markers. Instead, we added a figure to the SI (Fig. S7) where the 2h samples are distinguished by color from the 8 h samples.

Fig 6: What does it mean by "stands out in terms of ice activity"? Was this qualitatively or quantitatively determined? Either way, there should be some description of how this was defined.
This was a qualitative categorization that meant that the samples belong to one of the clusters described in the text. However, with the addition of the LINA measurements to Fig. 5 it was not possible anymore to indicate the individual clusters without overcrowding the figure. In the revised version we divide the area with the majority of very similar samples and those that we consider standing out by line

Fig 11: Clarify in the caption that the grey spectra are those from the entire study.
Added to the caption.

References
Conen, F., Eckhardt, S., Gundersen, H., Stohl, A., Yttri, K.E., 2017. Rainfall drives atmospheric ice-nucleating particles in the coastal climate of southern Norway. Atmos. Chem. Phys. 17, 11065–11073. https://doi.org/10.5194/acp-17-11065- 2017.

Conen, F., Henne, S., Morris, C.E., Alewell, C., 2012. Atmospheric ice nucleators active ≥–12 °C can be quantified on PM10 filters. Atmos. Meas. Tech. 5, 321–327. https:// doi.org/10.5194/amt-5-321-2012.

Conen, F., Yakutin, M.V., 2018. Soils rich in biological ice-nucleating particles abound in ice-nucleating macromolecules likely produced by fungi. Biogeosciences 15 (14), 4381–4385. https://doi.org/10.5194/bg-15-4381-2018

Conen, F., Morris, C.E., Leifeld, J., Yakutin, M.V., Alewell, C., 2011. Biological residues define the ice nucleation properties of soil dust. Atmos. Chem. Phys. 11, 9643–9648. https://doi.org/10.5194/acp-11-9643-2011.

Felgitsch, L., Baloh, P., Burkart, J., Mayr, M., Momken, M.E., Seifried, T.M., Winkler, P., Schmale Iii, D.G., Grothe, H., 2018. Birch leaves and branches as a source of ice- nucleating macromolecules. Atmos. Chem. Phys. 18, 16063–16079. https://doi.org/ 10.5194/acp-18-16063-2018.

Hara, K., Maki, T., Kakikawa, M., Kobayashi, F., Matsuki, A., 2016a. Effects of different temperature treatments on biological ice nuclei in snow samples. Atmos. Environ. 140, 415–419. https://doi.org/10.1016/j.atmosenv.2016.06.011.

Hill, T. C. J., DeMott, P. J., Tobo, Y., Fröhlich-Nowoisky, J., Moffett, B. F., Franc, G. D., and Kreidenweis, S. M.: Sources of organic ice nucleating particles in soils, Atmos. Chem. Phys., 16, 7195–7211, https://doi.org/10.5194/acp-16-7195-2016, 2016.

Huang, S., Hu, W., Chen, J., Wu, Z., Zhang, D. and Fu, P., 2021. Overview of biological ice nucleating particles in the atmosphere. Environment International, 146. https://doi.org/10.1016/j.envint.2020.106197.

Irish, V. E., Hanna, S. J., Willis, M. D., China, S., Thomas, J. L., Wentzell, J. J. B., Cirisan, A., Si, M., Leaitch, W. R., Murphy, J. G., Abbatt, J. P. D., Laskin, A., Girard, E., and Bertram, A. K.: Ice nucleating particles in the marine boundary layer in the Canadian Arctic during summer 2014, Atmos. Chem. Phys., 19, 1027–1039, https://doi.org/10.5194/acp-19-1027-2019, 2019.

Joly, M., Amato, P., Deguillaume, L., Monier, M., Hoose, C., Delort, A.M., 2014. Quantification of ice nuclei active at near 0°C temperatures in low-altitude clouds at the Puy de Dôme atmospheric station. Atmos. Chem. Phys. 14, 8185–8195. https:// doi.org/10.5194/acp-14-8185-2014.

Kunert, A.T., Pöhlker, M.L., Tang, K., Krevert, C.S., Wieder, C., Speth, K.R., Hanson, L.E., Morris, C.E., Schmale Iii, D.G., Pöschl, U., Fröhlich-Nowoisky, J., 2019. Macromolecular fungal ice nuclei in Fusarium: effects of physical and chemical processing. Biogeosciences 16, 4647–4659. https://doi.org/10.5194/bg-16-4647-2019.

McCluskey, C.S., Hill, T.C.J., Sultana, C.M., Laskina, O., Trueblood, J., Santander, M.V., Beall, C.M., Michaud, J.M., Kreidenweis, S.M., Prather, K.A., Grassian, V., DeMott, P. J., 2018. A Mesocosm Double Feature: Insights into the Chemical Makeup of Marine Ice Nucleating Particles. J. Atmos. Sci. 75, 2405–2423. https://doi.org/10.1175/ JAS-D-17-0155.1.

Moffett, B., Hill, T., DeMott, P., 2018. Abundance of Biological Ice Nucleating Particles in the Mississippi and Its Major Tributaries. Atmosphere 9, 307. https://doi.org/ 10.3390/atmos9080307.

Pouleur, S., Richard, C., Martin, J. G., & Antoun, H. (1992). Ice nucleation activity in Fusarium acuminatum and Fusarium avenaceum. Applied and environmental microbiology, 58(9), 2960-2964.

Sanchez-Marroquin, A., Arnalds, O., Baustian-Dorsi, K. J., Browse, J., Dagsson-Waldhauserova, P., Harrison, A. D., Maters, E. C., Pringle,K. J., Vergara-Temprado, J., Burke, I. T., McQuaid, J. B., Carslaw, K. S., and Murray, B. J.: Iceland is an episodic source of atmospheric ice-nucleating particles relevant for mixed-phase clouds, Science Advances, 6, https://doi.org/10.1126/sciadv.aba8137, 2020

U.S. EPA, Guidance for data quality assessment. Practical methods for data analysis. (Office of Environmental Information, Washington DC, 2000).

Welti, A., Bigg, E. K., DeMott, P. J., Gong, X., Hartmann, M., Harvey, M., Henning, S., Herenz, P., Hill, T. C. J., Hornblow, B., Leck, C., Löffler, M., McCluskey, C. S., Rauker, A. M., Schmale, J., Tatzelt, C., van Pinxteren, M., and Stratmann, F.: Ship-based measurements of ice nuclei concentrations over the Arctic, Atlantic, Pacific and Southern oceans, Atmos. Chem. Phys., 20, 15191–15206, https://doi.org/10.5194/acp-20-15191-2020, 2020.

Zeppenfeld, S., van Pinxteren, M., Hartmann, M., Bracher, A., Stratmann, F., and Herrmann, H.: Glucose as a potential chemical marker for ice nucleating activity in Arctic seawater and melt pond samples, Environmental Science & Technology, p. acs.est.9b01469,https://doi.org/10.1021/acs.est.9b01469, 2019

Zwaaftink, C. D. G., Grythe, H., Skov, H., and Stohl, A.: Substantial contribution of northern high-latitude sources to mineral dust in the Arctic, J. Geophys. Res.-Atmos., 121, 13678–13697, https://doi.org/10.1002/2016jd025482, 2016.

---

## Author Comment (AC2) · 5 Apr 2021

We thank referee 2 for reviewing our manuscript and for giving comments and suggestions. Our answers are given in blue, below, while the original text of the review was kept in black.

Referee #2 Evaluations:
**Scientific significance**: Excellent
**Scientific quality**: Excellent
**Presentation quality**: Excellent

General Comments

- The paper does and can only answer the INP source question in a very limited way. Even though a lot of different measurements/analysis is used, the conclusions are a bit vague, which is probably due to the complex nature of the research question. They find indications that INP above -15◦C are dominantly biogenic, while at lower temperatures dominantly dust (this is not so surprising looking at previous field studies, even though not so many specific studies for the Arctic exist). They state that the sources could be marine, locally emitted for the INP above -15◦C, but only if the INP are enriched during aerosolisation by several orders of magnitude. It would be helpful to discuss more critically the limitation of the study, what can be taken from it for which conditions and what needs further investigation. The authors might need to break down the complex research question into subquestions etc..

We are aware that a campaign-based study such as ours has several limitations, which is why we specified the time of year and location in the title of the paper. To better highlight this restriction we now also explicitly mention this limitation directly after formulating our research questions: "Our findings are constrained to the season and region in in which measurements took place. They will nevertheless contribute to a better understanding concerning Arctic INPs and their potential effects on Arctic clouds." (L81 – L82)

- Why are two different filter techniques (LINA and INDA) used and what are the advantages/disadvantages of both compared to each other? When is which technique used (figure captions are not always clear on that?)?

The main difference between LINA and INDA is that they are sensitive in different, but overlapping ranges of INP concentration (as described in section 2.3.5). In our eyes, the difference in their respective detectable range is neither an advantage or disadvantage but give a more complete dataset when combined.

The question which instrument was used was also asked by referee 1, therefore we repeat our answer given there: Which instrument was used was constrained by instrument availability. While the filter samples were scheduled to be analyzed, INDA was not continuously available, hence a full dataset for these exists only from LINA. For the sea water samples LINA and INDA measurements are available, and are now also shown in Fig, 5. In general, INDA measurements are more sensitive to low INP concentrations at warm temperatures that are relevant for the biological INP and also provide better signal to background ratio (sample in comparison to pure MilliQ water). Hence, data measured with INDA was preferred over LINA were possible.

In the revised version of the manuscript, we now indicate which instrument was used in the figures showing freezing spectra (Fig. 2, 3, 5, 6, 7, 8, 10 and 11).

- It is not clear where the conclusions come from that the lower temperature INP are

dominated by dust (no clear indication from the trajectories?). Is that derived from the slope of the measurement and comparison to other field measurements?

*Please note that a referee 1 made a comment on the same topic, hence some parts of our answer here are identical to our answer given there (p. 7 there).*

This stems primarily from the abundance and the temperature range of the INP we observed in comparison with other field measurements: To our knowledge, no study has yet found biologic INP to be present in such high numbers in that temperature range as shown in our Fig. 4. Also, only mineral dust is known to occur in such high atmospheric concentrations that it could explain the observed INP concentrations. The study by Welti et al. (2020) does confirm this assumption as they only observe similarly high INP concentrations at the same temperature, when they are in the outflow region of the Sahara, where the aerosol is known to be dominated by mineral dust. Additionally the study by Groot Zwaaftink et al. (2016) shows that mineral dust is produced within the Arctic, which is supported by the recent study by Sanchez-Marroquin et al. (2020), where Icelandic dust is identified as an important source for INP in the lower temperature regime. Then also Irish et al. (2019a) suggested that observed INP were mineral dust particles originating in the Arctic (Hudson Bay, eastern Greenland, northwest continental Canada), rather than particles originating from sea spray. These are of course only indications, but we also don't see any reason to assume that high concentrations of non-mineral INP that are ice active at low temperature should occur especially in the Arctic.

The whole paragraph now reads as follows: "In the previous section we described that at warmer temperatures for example at -10°C, samples with high INP concentrations are found more often in the MIZ and less frequently within the ice pack. In comparison, at the lower temperatures measured with SPIN (cross markers in Fig. 4) no correlation with the environmental setting is found. However, in global context the level of $N_{INP}$ at these low temperatures is remarkable by itself as shown in Fig. 4. That figure shows $N_{INP}$ in the Arctic at -32°C measured with SPIN during PS106, but also SPIN data by Welti et al. (2020) of a transect from Bremerhaven (Germany) to Cape Town (South Africa) along the western coast of Africa. It is striking that at these low temperatures $N_{INP}$ in the Arctic are in the same order of magnitude as in the outflow region of mineral dust from the Saharan desert. While we have no means of proofing the presence of mineral dust at these colder temperatures during PASCAL, to our knowledge there are also no other known sources of INP that can produce such high concentrations throughout whole time period of the campaign. Also, it was recently shown by Sanchez-Marroquin et al. (2020) that Iceland can be a strong Arctic dust source. Also Irish et al. (2019a) suggested that observed INP were mineral dust particles originating in the Arctic (Hudson Bay, eastern Greenland, northwest continental Canada), rather than particles originating from sea spray. And global model transport simulations done by Groot Zwaaftink et al. (2016) show that mineral dust is not only transported into the Arctic from remote regions but also, possibly increasingly, generated in the region itself. However, it is also possible also other sources of mineral INP contribute to the INP population at these temperatures. E.g., diatoms represent a biogenic, but mineral source of INP, as they have a cell wall made of silica (Xi et al., 2021) Therefore, it is likely that mineral INP, possibly mineral dust, contribute to $N_{INP}$ at low temperatures during the campaign." (L311-327)

- Section 3.3.1: How large is the variation due to the assumptions made (SS etc.)? Could you add a section on commenting this and or add error ranges depending resulting from the assumptions made? Same for the variable N_CCN which might affect the result (and comes itself already with an uncertainty).

We have no measures of the actual SS during the fog events we encountered during the cruise. Therefore, the only way to assess the variation of the fog water derived INP concentrations is to vary $N_{CCN}$ at different SS. As mentioned in the manuscript, the literature values for SS of fog range from 0.02% to 0.2%. Since no $N_{CCN}$ below a SS of 0.1% were measured, we calculate an estimate for $N_{CCN}$ at 0.02%SS as follows: We derived the factor by which $N_{CCN}$ decreases from 0.3%SS to 0.2%SS (per 0.1% decrease in SS, $N_{CCN}$ decreases by a factor of 0.81). With that factor we estimated $N_{CCN}$ at 0.02%SS. Then, we calculated again the INP concentration as described in the main manuscript. A linear extrapolation to such low supersaturations has large uncertainties but gives an estimate for the lower boundary of the presented $N_{INP}$ derivation. While the agreement between the scaled down fog water derived INP concentrations and the INP in the air (measured on filter samples) is not as good it was before, the majority of the fog water samples still overlap by one to almost two orders of magnitude. We see still a general agreement between the INP concentration in the air and the fog water

derived INP concentration, especially if the simplicity of this closure calculation is considered. We included a comment on this uncertainty in section 3.3.1 (L428-432): "In the SI (section S9), the fog water derived $N_{INP}$ are shown for an extrapolated value of $N_{CCN}$ at SS = 0.02%. With that value, the agreement between the filter and fog derived $N_{INP}$ is reduced, nevertheless both still overlap by one to almost two orders of magnitude. A linear extrapolation to such low supersaturations has large uncertainties, hence it should be only seen as an estimate for the lower boundary of the presented derivation method of $N_{INP}$ in air from fog water samples."
Also the the calculation and figure presented above are added to the SI (section S9).

- How large are the uncertainties in the calculated trajectories? Please comment on limitations here as well.
The uncertainty for the 3-day back trajectories shown in Fig.13 are ca. 260 km. This uncertainty was derived by arranging an ensemble of 4 trajectories around the receptor site (i.e., the location of the vessel) with a distance of 0.05°. Then the average distance of the ensemble members to the center trajectory was calculated.
Additional uncertainty stems from the resolution of the meteorological input file given to the HYSPLIT model. We used the standard input for HYSPLIT (GFS1), which as 1° resolution (corresponding to

111km in latitude and 23km in longitude for the area around Svalbard). Hence this is the inherent uncertainty at any point along any trajectory we show. Kahl (1993) described high uncertainties (ca. 1000 km) for 5-day back trajectories in the Arctic. Therefore, we have refrained from interpreting back trajectories that extend beyond three days. We consider the trajectories as only general pointer towards a source region from which an air mass originates. Nothing added.

- Page 2, line 21 and 22: The sentence on the heat induced reduction... can be re-moved, this was stated already on page 1, line 9 ff.
Sentence was removed.

- Page 2, line 35-46: It would make sense to switch paragraph line 35-39 with the paragraph line 40-46.
Paragraphs were switched.

- Page 5, line 127: "within the ice pack" = open leads?
Yes, in the open leads within the ice pack. This is now specified in the manuscript: "[…] Seawater samples were taken from different environments, i.e., ice-free ocean, marginal ice zone (MIZ), open leads within the ice pack or from meltponds.[…] " (L126-L127)

- Page 9, line 246: What is a "vessel's Ferrybox system"? How does it work?
A Ferrybox is an online instrument to continuously measure oceanographic parameters in a flow-through system that was developed in the early 2000s and is commercially available. A Ferrybox system consists of a water inlet from which seawater is pumped into the measuring circuit containing multiple, modular sensors. Ferryboxes are deployed on research vessels as well as ferries, cruise, and cargo ships (see Petersen et al. 2007, 2011, for details). In the case of the RV Polarstern, the seawater inlet is located in a depth of around 11m, but due to the water flow around the body of the ship, also water from above can get in.

In the revised version the respective text passage now reads as (L260-263): "Chlorophyll-a (Chl-a) concentration was derived from the vessel's Ferrybox system (4H-FerryBox, Jena Engineering, Jena, Germany). A ferrybox is an autonomous online instrument with modular sensor assembly to continuously measure oceanographic parameters in a flow-through system (Petersen et al. 2011, Petersen et al. 2007). The data from the Chl-a sensor in the Ferrybox system were accessed via the DSHIP portal (https://dship.awi.de/) provided by the operator of the vessel."

- Page 12, line 290-291: Was the sampling time consistent for all environmental set-tings or how can that influence the statement?
The sampling time was set consistently (automatic sample changer) for 8h throughout the campaign except for four days, when the sampling period was reduced to 2h (see section 2.2.1). The statement is meant in a relative sense, i.e. that a higher fraction of the samples collected in the MIZ and ice-free ocean environment are show higher INP concentrations at the warmer temperatures compared to samples from within the ice pack environment. Since the statement does not refer to the absolute number of samples, but the relative number, the sampling time should not significantly influence the metric. In the revised version the statement is changed to:
"[…] at warmer temperatures for example at -10°C, samples with high INP concentrations are found more often in the MIZ and less frequently within the ice pack. […]" (L304-305)

- Fig. 4: Is that figure needed in this paper?

In response to referee #1 additional discussion of the SPIN data was added in the revised version of the manuscript, which is why we would prefer to keep this figure in the manuscript. The Figure shows the surprisingly high INP concentration in the Arctic environment, concentrations which have previously only been observed within the Saharan dust outflow.

- Fig. 7: The labels are a bit small here.
Figure labels are larger in the revised manuscript.

- Page 15, line 340: Add: "...can be seen" due to filtration.
Suggestion added.

- Page 16, line 354-355: Is it legitimate to compare sea water and fog water directly?
Is there not a bias due to dillution effects?
We agree with the comment that this is not legitimate and removed the respective sentence.

- Fig. 11 a: The fit/slope at lower T is very difficult to spot in the graph.
The lines are now wider in the revised version.

- Fig. 12: Write variables in the caption.
Variables were added.

- Fig. 12: The arrows are very small to spot.
We assume the comment refers to panel b of that figure and enlarged the arrows there.

- Page 24, line 503: Be more specific what "similar in shape" means.
With "similar shape" we refer to the steep slope between ca. -7°C and -10°C, followed by an extended plateau region until ca. -21°C that leads into less steep slopes compared to warmer temperatures. A shortened version of this description is added to the bullet point to which the referee referred to: "The freezing spectra of atmospheric INPs are similar in shape, i.e. a steep slope at warm temperatures followed by an extended plateau region, followed by a less steep slope, indicating that during the case study consistent atmospheric INP populations were sampled." (L540-542)

- Page 25, line 537-538: The most enriched samples featured the highest ice activity.
-> that refers to the SML samples?
Yes. This was made clearer in the revised version (L576-577): "The most enriched samples featured the highest ice activity in the SML samples."

Technical corrections:
- Page 11, line 270: Remove brackets around the temperatures.
- Page 14, Caption Fig. 5: ...the samples were taken from.
- Page 14, Caption Fig. 5: Add a . at the end of the caption.
- Page 20, line 432: Remove the brackets around the citation.
All technical corrections were implemented in the revised manuscript.

References

Kahl, J. D. (1993). A cautionary note on the use of air trajectories in interpreting atmospheric chemistry measurements. *Atmospheric Environment. Part A. General Topics*, *27*(17-18), 3037-3038.

Petersen W et al. (2007) FerryBox: from online oceanographic observations to environmental information. In: Petersen W, Colijn F, Hydes D, Schroeder F (eds) EuroGOOS Publication No. 25. EuroGOOS Office, SHMI, 601 76 Norkoepping, Sweden ISBN978-91097828-4-4

Petersen, W., Schroeder, F. & Bockelmann, FD. FerryBox - Application of continuous water quality observations along transects in the North Sea. *Ocean Dynamics* **61,** 1541–1554 (2011). https://doi.org/10.1007/s10236-011-0445-0

Welti, A., Bigg, E. K., DeMott, P. J., Gong, X., Hartmann, M., Harvey, M., ... & Stratmann, F. (2020). Ship-based measurements of ice nuclei concentrations over the Arctic, Atlantic, Pacific and Southern oceans. *Atmospheric Chemistry and Physics*, *20*(23), 15191-15206.

---

## Author Response (AR2)

We thank referee #1 for reviewing our manuscript and for giving comments and suggestions. Our answers are given in blue, below, while the original text of the review was kept in black.

Referee #1 Evaluations:
**Scientific significance**: Excellent
**Scientific quality**: Excellent
**Presentation quality**:  Excellent

I have no additional major comments. The authors did a nice job of addressing the first reviews. My only technical suggestion would be to lighten the gridlines in the figures. There are many colors and shapes for the spectra (which cannot be avoided given the amount of data that need to be represented), but the gridlines somewhat draw attention away from the spectra themselves and make the plots a bit busy. Perhaps making them dashed and/or a lighter grey color would be less distracting and emphasize the spectra.

The gridlines in the spectra plots were already dotted. However, in the revised version, we changed their thickness from 0.5 pt to 0.3 pt and also made them slightly lighter in color (Figures 3, 5, 6, 8, 9, and 11).

We thank referee #3, Russel Schnell, for reviewing our manuscript and for giving comments and suggestions. Our answers are given in blue, below, while the original text of the review was kept in black.

Referee #3 Evaluations:
**Scientific significance**: Excellent
**Scientific quality**: Excellent
**Presentation quality**: Excellent

Line 610: The authors comment on the high concentration of organic INP in the marine atmosphere compared to those in the ocean waters. They suggest that enhancements of marine organic material between the marine concentrations and atmospheric concentrations would have to be several orders of magnitude to explain the observed atmospheric concentrations.

In fact, such enhancements are well documented to explain the "orders of magnitude" the authors state is required to explain their observations. What they have observed is a phenomenon that was studied in the 1960s and 1970s related to marine bubble bursting jets that highly concentrate surface microlayer organic material and bacteria and eject them into the atmosphere.

The authors should become familiar with the publications of D.C. Blanchard. One that addresses the enhancement of organics and bacteria between marine waters and air is:

"Jet drop enrichment of bacteria, virus, and dissolved organic material, Duncan C. Blanchard, Pure and Applied Geophysics, Volume 116, pages 302–308 (1978).

He and his associated have other papers on the subject.

I suggest the manuscript should be modified to reference and discuss the earlier measurements on the enhancement process. This addition will nicely address one of the unanswered items posed by the authors enhance the veracity of their measurements. And their courage to present observations that some might construe as faulty measurements as the results seem counterintuitive.

With this small modification, I suggest this important paper is ready for publication and that it will be well received and referenced many times as it contains such a wealth of important Arctic relevant information.
In the revised manuscript, we added a paragraph that describes the findings by Blanchard (1978) as suggested by the referee:
"However, studies about the enrichment of bacteria and organic matter exist. Blanchard (1978) describes that in jet drops, which are produced when bubbles burst at the air-water interface, bacteria can get enriched by a factor greater than $10^3$. While several factors, including the type of bacteria themselves, control the EF of bacteria, the findings by Blanchard (1978) suggest that similar EFs may also apply to INP, since bacteria are a major contributor to seawater ice activity, as described in the introduction." (L462-L466)

In the summary, where the results of the closure calculations are summarized, the findings presented in Blanchard (1978) are taken up again: "However, literature suggests that such EFs greater than $10^3$ may be possible for INP." (L591-L592)

Note to the authors: If this was my paper, I would change the title to read something along the like of "Are all the most active terrestrial and marine ice nuclei observed during the ... of biological origin?"
We thank the reviewer for the suggestion, but the authors prefer to retain the original title.